# Promoting nickel oxidation state transitions in single-layer NiFeB hydroxide nanosheets for efficient oxygen evolution

Yuke Bai[1,5], Yu Wu[2,5], Xichen Zhou ●[1], Yifan Ye[3], Kaiqi Nie[4], Jiaou Wang ●[4], Miao Xie ●[2], Zhixue Zhang[1], Zhaojun Liu[1], Tao Cheng ●[2] ✉ & Chuanbo Gao ●[1] ✉

Promoting the formation of high-oxidation-state transition metal species in a hydroxide catalyst may improve its catalytic activity in the oxygen evolution reaction, which remains difficult to achieve with current synthetic strategies. Herein, we present a synthesis of single-layer NiFeB hydroxide nanosheets and demonstrate the efficacy of electron-deficient boron in promoting the formation of high-oxidation-state Ni for improved oxygen evolution activity. Raman spectroscopy, X-ray absorption spectroscopy, and electrochemical analyses show that incorporation of B into a NiFe hydroxide causes a cathodic shift of the $Ni^{2+}(OH)_2 \rightarrow Ni^{3+\delta}OOH$ transition potential. Density functional theory calculations suggest an elevated oxidation state for Ni and decreased energy barriers for the reaction with the NiFeB hydroxide catalyst. Consequently, a current density of $100\ mA\ cm^{-2}$ was achieved in 1 M KOH at an overpotential of 252 mV, placing it among the best Ni-based catalysts for this reaction. This work opens new opportunities in electronic engineering of metal hydroxides (or oxides) for efficient oxygen evolution in water-splitting applications.

Electrochemical water splitting is one of the most promising routes for mass production of ultrapure hydrogen for applications such as fuel cells[1–4]. However, its efficiency is largely impeded by the oxygen evolution reaction (OER) occurring at the anode because this is a sluggish process involving 4 proton-coupled electron-transfer steps (PCETs)[5–7]. To date, the most successful commercial catalysts for the OER are noble metal oxides, such as $IrO_2$ and $RuO_2$[8,9]. To address the scarcity of noble metals, earth-abundant nonnoble transition-metal-based catalysts, such as NiFe[10–16] and Co-based[17,18] oxides or hydroxides, have been widely explored as alternative catalysts. In particular, state-of-the-art NiFe-layered double hydroxides (NiFe-LDHs) have shown excellent activity in the OER under alkaline conditions[4,19–22]. During the OER, the transition metals (e.g., $Ni^{2+}$) evolve into high-oxidation-state species (i.e., NiOOH with the oxidation state of Ni greater than +3), which are recognized as active sites for this reaction[23–28]. Therefore, we expect that promoting the formation of the desired high-oxidation-state transition metal species may lead to a decreased onset potential and therefore enhanced kinetics for the OER.

To date, great effort has been devoted to developing highly efficient Ni-based catalysts for the OER. However, research on consciously increasing the oxidation state of Ni to boost OER activity has been rather limited. When a Ni-based catalyst was conjugated to plasmonic nanoparticles, positively charged "holes" were produced in the nanoparticles under visible light irradiation, which readily enhanced oxidation of nearby $Ni^{2+}$ into high-oxidation-state species, showing improved OER activity[29,30]. Conjugation of Ni oxide catalysts with other

[1]State Key Laboratory of Multiphase Flow in Power Engineering, Frontier Institute of Science and Technology, Xi'an Jiaotong University, Xi'an, Shaanxi 710054, China. [2]Institute of Functional Nano & Soft Materials (FUNSOM), Jiangsu Key Laboratory for Carbon-Based Functional Materials & Devices, Joint International Research Laboratory of Carbon-Based Functional Materials and Devices, Soochow University, Suzhou, Jiangsu 215123, China. [3]National Synchrotron Radiation Laboratory, University of Science and Technology of China, Hefei 230029, China. [4]Institute of High Energy Physics, Chinese Academy of Sciences, Beijing 100049, China. [5]These authors contributed equally: Yuke Bai, Yu Wu. ✉e-mail: tcheng@suda.edu.cn; gaochuanbo@mail.xjtu.edu.cn

metal oxides, such as $MoO_2$ and $WO_x$, and nonmetals, such as phosphate, was also found to be effective in forming high-oxidation-state Ni for the OER[28,31]. As a result, OER activities can be greatly enhanced with these catalysts. Despite the encouraging features in these prior endeavors, the OER activities were still far from optimal. It remains highly desirable, albeit challenging, to explore effective strategies for promoting oxidation of Ni to push the OER activity to unprecedented levels.

Herein, we show that oxidation state transition of Ni can be effectively promoted by introducing electron-deficient boron (B) into conventional NiFe hydroxide catalysts, which significantly improves the OER activity. Usually, direct extraction of electrons from $Ni^{2+}$ to form the desired $Ni^{3+\delta}$ is a difficult process and can only be achieved at a high potential. We expect that B in the proximity of Ni may participate in Ni oxidation by serving as a transit site for the electron flow involved in this process. Owing to its intrinsic electron deficiency, the transit B site may act as an "electron sink" to facilitate electron flow from the $Ni^{2+}$ site, thus allowing formation of active $Ni^{3+\delta}$ at a reduced potential, which is beneficial for the OER[29]. To verify this idea, we synthesized single-layer NiFeB hydroxide nanosheets via in situ hydrolysis of NiFeB alloy nanoparticles. Raman spectroscopy, X-ray absorption spectroscopy, and electrochemical measurements showed that the potential for conversion of $Ni^{2+}(OH)_2$ to $Ni^{3+\delta}OOH$ in the NiFeB hydroxide nanosheets was lower than that seen with the B-free NiFe hydroxide nanosheets used for reference. Density functional theory (DFT) calculations showed that the Ni atom showed a higher oxidation state upon incorporating B, suggesting electronic interactions between the Ni and the B components. As a result, the OER activity of the NiFeB hydroxide nanosheets reached $100 \, mA \, cm^{-2}$ at an overpotential of 252 mV, which was superior to the OER activities of B-free NiFe hydroxide nanosheets (overpotential, 337 mV) and most Ni-based catalysts reported to date. Although borate has been used as an additive to the electrolyte[25,27,32] and transition metal borides have been used directly as catalysts for the OER[33–43], to the best of our knowledge, the effect of B remains unclear. Our work reveals an unambiguous effect of the electron deficiency of B in promoting the oxidation state transition of Ni, which provides new opportunities for designing advanced metal hydroxide (or oxide) catalysts for efficient OER toward improved viability of electrochemical water-splitting applications.

## Results

### Synthesis and characterization

Single-layer NiFeB hydroxide nanosheets were synthesized by in situ hydrolysis of NiFeB alloy nanoparticles under ambient conditions (Fig. 1a). First, NiFeB alloy nanoparticles were synthesized by reducing $Ni(NO_3)_2$ and $Fe(NO_3)_3$ with $NaBH_4$ in an ethanolic solution under sonication. $NaBH_4$ served as both a reducing agent and a boron source. Transmission electron microscopy (TEM) showed that the product was composed of nanoparticles with an average size of ~4 nm (Figs. S1 and S2 show a control sample without B). X-ray photoelectron spectroscopy (XPS) confirmed the presence of Ni, Fe, and B in the nanoparticles (Fig. S3). The profiles of Ni, Fe, and B were well fitted by peaks for oxidized ($Ni^{2+}$, $Fe^{3+}$, and B–O) and zero-valent species ($Ni^0$, $Fe^0$, and $B^0$). The presence of zero-valent Ni, Fe, and B clearly confirmed formation of a NiFeB alloy after the chemical reduction. The oxidized species was attributed to oxidation of zero-valent species on the surface by ambient air. Next, the NiFeB alloy nanoparticles were converted into hydroxides by dispersing the nanoparticles in an alkaline solution, i.e., KOH. The NiFeB alloy nanoparticles underwent quick hydrolysis and evolved into ultrathin nanosheets (TEM image, Fig. 1b). These nanosheets were tangled into bundles with abundant wrinkles. The thicknesses of the nanosheets were measured by TEM imaging of a wrinkle that stood vertical to the carbon support on the Cu grid. The thickness of the wrinkle was measured to be ~1 nm, which corresponded to 2 layers of nanosheets (Fig. 1b, inset). Thus, the thickness

of a single-layer nanosheet was ~0.5 nm. Energy-dispersive X-ray spectroscopy (EDS) elemental mapping showed that Ni, Fe, and B were uniformly distributed in the nanosheets and formed homogeneous solid solution rather than segregated phases (Fig. 1c). X-ray photoelectron spectroscopy (XPS) revealed that zero-valent species disappeared after hydrolysis and showed only signals for oxidized species ($Ni^{2+}$, $Fe^{3+}$, and B–O); this suggested that the nanosheets were composed of a NiFeB hydroxide, i.e., cationic Ni, Fe, B connected by bridging O or OH (Figs. 1d and S3). B $K$-edge X-ray absorption spectroscopy (XAS) also showed a sharp peak with a photon energy of 194 eV, which corresponded to the electron transition in a $BO_3$ group with a trigonal configuration (Fig. S4), which was consistent with the XPS results[44].

We further identified the crystalline structure of the ultrathin nanosheets. The X-ray diffraction (XRD) pattern showed reflections at ~34° and 60° ($2\theta$), which corresponded to reflections from the 100 and 110 planes of edge-sharing $MO_6$ (M = Ni, Fe) octahedra in a metal hydroxide, similar to the structure observed in NiFe-LDHs (Fig. 1e)[45]. No other reflection peaks were observed, particularly those from stacked edge-sharing $MO_6$ planes, suggesting that the nanosheets were composed of a single $MO_6$ layer. According to previous reports, the thickness of an edge-sharing $MO_6$ plane is ~0.48 nm[4,46–48], which perfectly matched the thickness of the nanosheets as measured by TEM (~0.5 nm), again confirming the single-layer structure of the nanosheets. Typically, in NiFe-LDH, the positive charges carried by $Fe^{3+}$ are balanced by free anions. To exclude the presence of B adsorbed as a borate ion on the surface, we conducted an anion exchange reaction of the nanosheets with $0.25 \, M \, NaNO_3$ for 18 h. The B content in the nanosheets was then examined with inductively coupled plasma–mass spectrometry (ICP–MS) (Table S1). No loss of B was detected, confirming that B was not present as a borate counterion adsorbed on the nanosheets but was incorporated into the nanosheets via B–O–Ni or B–O–Fe bridging bonds.

### Effect of B on the oxidation state transition of Ni

To reveal the effect of B, we collected Raman spectra of the NiFeB hydroxide nanosheets to monitor the oxidation state transitions of Ni occurring at different overpotentials (electrolyte: $O_2$-saturated 1.0 M KOH) (Fig. 2a). There were no obvious Raman signals in the spectrum of the as-synthesized NiFeB hydroxide nanosheets. When the overpotential was increased to 66 mV (i.e., 1.296 V vs. a reversible hydrogen electrode, RHE), well-resolved peaks appeared at ~474 and 558 $cm^{-1}$, which corresponded to $Ni^{3+\delta}OOH$[10,32,37,49,50]. These peaks were retained as the overpotential was increased to 166, 266, and 366 mV. The broad Raman peak at 680 $cm^{-1}$ can be attributed to Fe–O species in the NiFeB hydroxide nanosheets, as this peak has also been observed for $Fe_3O_4$, $\gamma$-$Fe_2O_3$, and $\gamma$-FeOOH, which contain the same structural unit[10]. These experimental observations proved that Ni in the NiFeB hydroxide nanosheets existed as $Ni^{3+\delta}OOH$ at all overpotentials investigated. The active $Ni^{3+\delta}OOH$ species formed in the NiFeB hydroxide nanosheets at overpotentials of 66 mV or higher.

The Raman spectra of ultrathin B-free NiFe hydroxide nanosheets at different overpotentials were investigated for comparison (Fig. 2b). The as-synthesized NiFe hydroxide nanosheets showed pronounced Raman peaks at 449 and 527 $cm^{-1}$, which were attributed to vibrations of $Ni^{2+}$–OH and $Ni^{2+}$–O bonds, respectively, in $Ni^{2+}(OH)_2$[10,32,37,49,50]. These peaks remained unchanged through the overpotential range of 0–166 mV, suggesting that Ni in the nanosheets remained as $Ni^{2+}(OH)_2$ under these conditions. These $Ni^{2+}(OH)_2$ species are inactive for the OER. The active Ni species for the OER, i.e., $Ni^{3+\delta}OOH$, appeared only when the overpotential was increased to 266 mV or higher (e.g., 366 mV). Therefore, formation of the $Ni^{3+\delta}OOH$ species active in the OER is a more difficult process in B-free NiFe hydroxide nanosheets.

It is worth noting that Fe contributed to the boosted oxidation state transition of Ni in the hydroxide nanosheets. Raman spectroscopy suggested that the overpotential required for the $Ni^{2+}(OH)_2 \rightarrow Ni^{3+\delta}OOH$

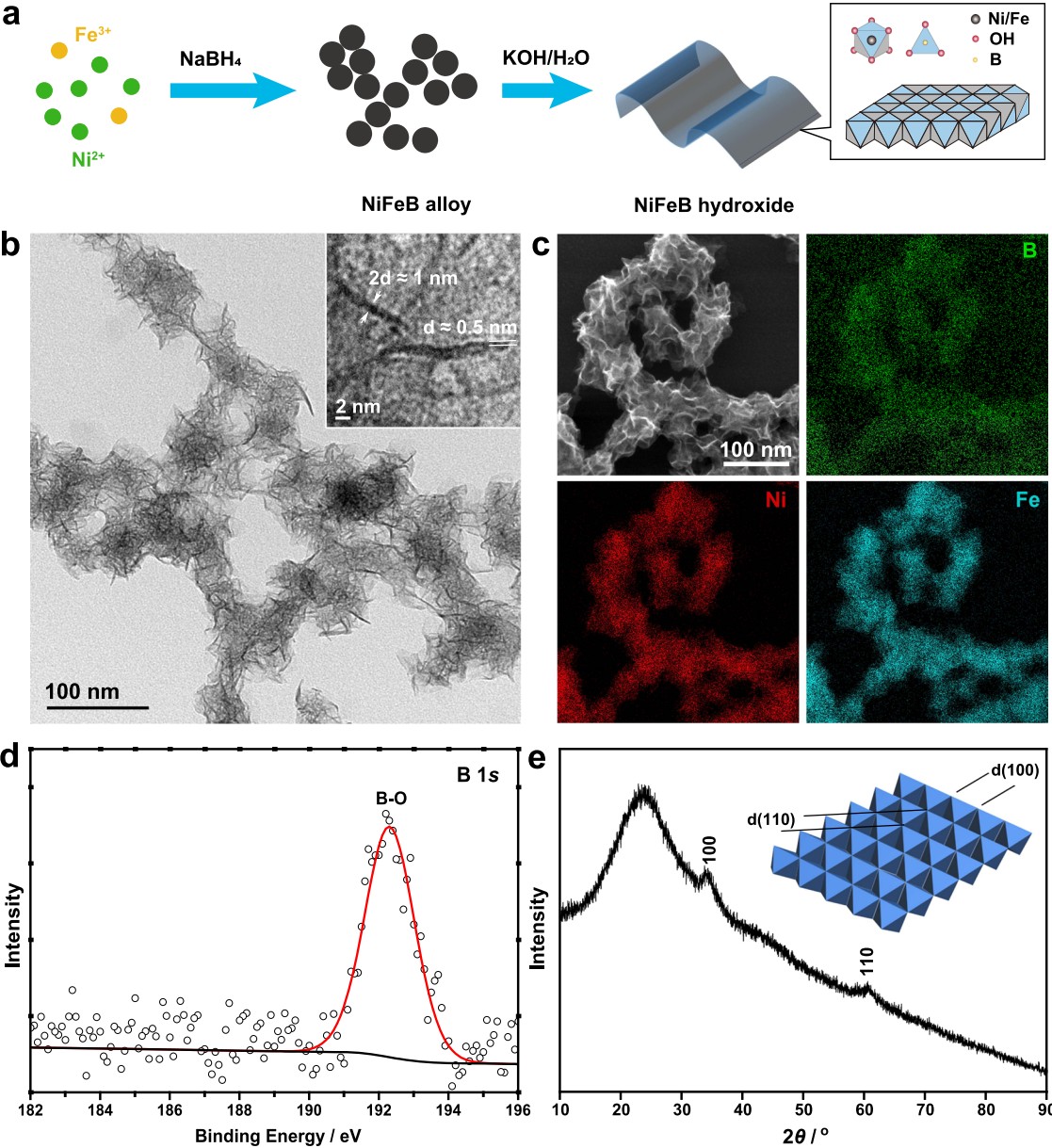

**Fig. 1 | Synthesis and characterization of single-layer NiFeB hydroxide nanosheets. a** Schematic synthetic procedures. **b** TEM image of the nanosheets. Inset: thickness of the nanosheets at the edge of a wrinkle. **c** EDS elemental maps for Ni, Fe, and B. **d**, **e** B 1$s$ XPS and XRD pattern of the NiFeB hydroxide nanosheets. Inset of **e**: a model of the crystal structure.

transition was 166 mV in an Fe-free NiB hydroxide catalyst (Fig. S5), which was higher than the overpotential required for a NiFeB hydroxide catalyst observed with the same technique (66 mV, Fig. 2a). Therefore, Fe in the hydroxide catalyst is indispensable for achieving high catalytic efficiency.

We further employed XAS to verify the oxidation state transition undergone by Ni in the NiFeB and NiFe hydroxide nanosheets in response to the applied overpotentials (Fig. 2c, d). The nanosheets were maintained at different overpotentials (66, 236, and 366 mV) prior to data collection. In the Ni $L_{3,2}$-edge XAS spectra, peaks i and iii arose from promotions of the electrons in a Ni 2$p$ orbital to a 3$d$ $t_{2g}$ orbitals in an octahedral crystal field, and peaks ii and iv arose from an electron transition from a Ni 2$p$ to a 3$d$ $e_g$ orbital. The ii/i peak intensity ratio reflects the ratio of holes in the $e_g$ and $t_{2g}$ orbitals, which is a measure of the oxidation state of Ni in the catalysts[28,51–53]. The as-synthesized NiFeB and NiFe hydroxide nanosheets showed low ii/i peak intensity ratios, corresponding to Ni²⁺. With increasing

overpotentials, the shape of the spectra changed dramatically with increases in the intensity of peak ii relative to that of peak i, suggesting a Ni oxidation state transition from +2 to +3 or higher. With the NiFeB hydroxide nanosheet catalyst, such an oxidation state transition resulted in the overpotential range 66–236 mV (Fig. 2c). For comparison, the oxidation state transition of Ni occurs at higher over-potentials of 236–366 mV with B-free NiFe hydroxide nanosheets (Fig. 2d). These results confirmed the critical role of B in promoting oxidation state transition of Ni, consistent with the trend shown by Raman spectroscopy.

Evolution of the Fe oxidation state in the NiFeB and NiFe hydroxide nanosheets at different overpotentials can also be revealed by Fe $L_{3,2}$-edge XAS spectroscopy (Fig. S6). For both catalysts, the spectra showed a more intense peak ii (Fe 2$p$ → 3$d$ $e_g$ transition) relative to peak i (Fe 2$p$ → 3$d$ $t_{2g}$ transition), which corresponded to Fe³⁺ in the catalysts[52,53]. The spectra displayed no significant changes with increasing overpotentials, suggesting that the oxidation state of Fe is

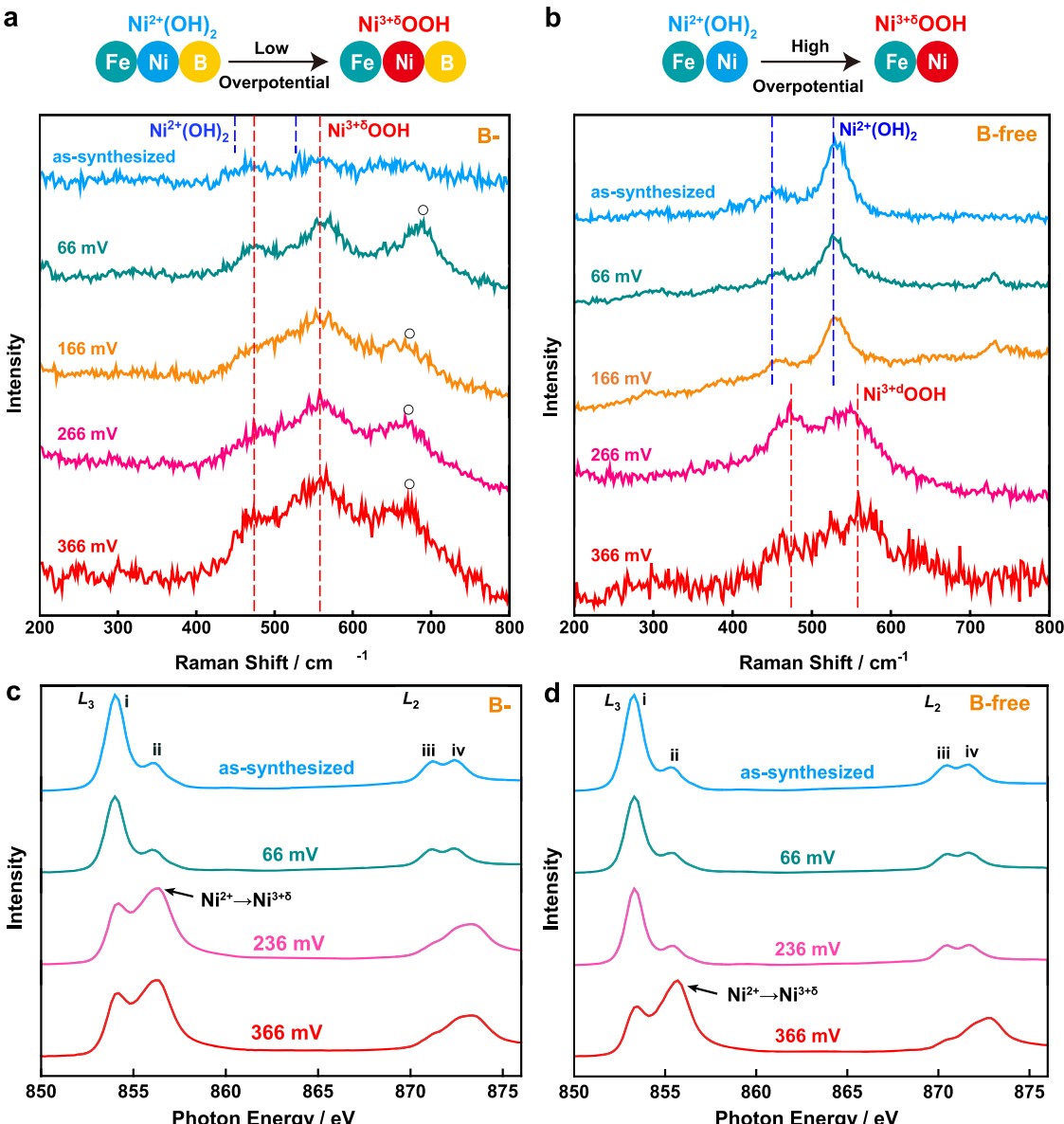

**Fig. 2 | Spectral investigations of the Ni$^{2+}$(OH)$_2$ → Ni$^{3+\delta}$OOH transitions in NiFeB and NiFe hydroxide nanosheets.** Raman spectra of the NiFeB hydroxide (**a**) and NiFe hydroxide nanosheets (**b**) after they were maintained at different overpotentials (66, 166, 266, and 366 mV). Blue and red dashed lines indicate the Raman shift positions for Ni$^{2+}$(OH)$_2$ and Ni$^{3+\delta}$OOH, respectively. Peaks labeled with circles in **a** were ascribed to Fe species. Top: Schematic showing conversions of Ni$^{2+}$(OH)$_2$ into Ni$^{3+\delta}$OOH at different potential thresholds. Ni $L_{3,2}$-edge XAS of the NiFeB hydroxide (**c**) and NiFe hydroxide nanosheets (**d**) after they were maintained at different overpotentials (66, 236, and 366 mV).

virtually constant during catalysis. Therefore, the active sites for the OER might be Ni$^{3+\delta}$OOH rather than Fe species. However, we cannot rule out the possibility of small numbers of Fe atoms on the surface evolving into even higher oxidation states during the OER, which are too reactive or too low in abundance to be observed by XAS analysis.

Both the Raman and XAS results verified the critical role of B in reducing the overpotential required for oxidation of Ni from inert Ni$^{2+}$ to active Ni$^{3+\delta}$, which promises improved activity of NiFeB hydroxide nanosheets in the electrocatalytic OER compared with the B-free counterpart.

**Electrocatalytic performance**

We then evaluated the OER performance of the NiFeB hydroxide nanosheets and compared it with that of B-free NiFe hydroxide nanosheets (Fig. 3). The nanosheets were supported on carbon and loaded on glassy carbon electrodes (RDEs) (loadings, Table S1). The OER activities were then measured by linear sweep voltammetry (LSV, anodic sweep from 1.043 to 1.743 V vs. RHE) at 10 mV s$^{-1}$ in O$_2$-saturated 1.0 M KOH, and the current densities were normalized to the geometric area of the electrode (Fig. 3a). The NiFeB hydroxide nanosheets showed significantly higher catalytic activity than the B-free NiFe hydroxide nanosheets. To reach a current density of 100 mA cm$_{geo}^{-2}$, the NiFeB hydroxide nanosheets required a much lower overpotential (252 mV) than NiFe hydroxide nanosheets (337 mV). The current density of the NiFeB hydroxide nanosheets at 1.53 V vs. RHE, i.e., with an overpotential of 300 mV, was 393 mA cm$_{geo}^{-2}$, which was 11.7 times greater than that of the NiFe hydroxide nanosheets (33.6 mA cm$_{geo}^{-2}$) (Fig. 3b). In terms of the current density produced at 1.5 V vs. RHE, the NiFeB hydroxide is a top-ranked catalyst for the OER compared with typical nonnoble metal catalysts reported in the literature (Fig. 3c and Table S2)[13,20,22,34,54]. It is worth noting that an oxidation peak appeared in both the LSV and cyclic voltammetric curves (CV, Fig. S7) of both

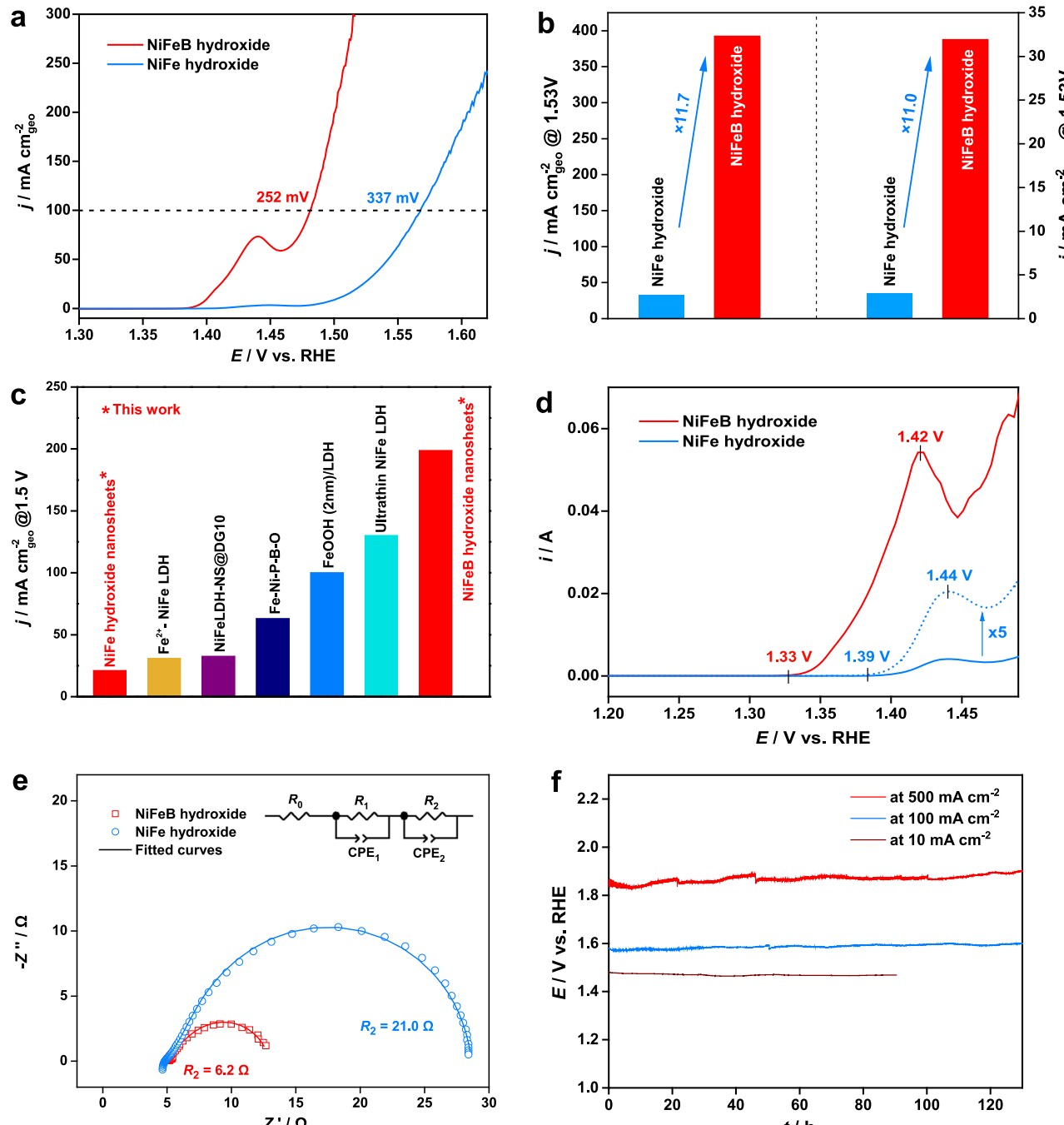

**Fig. 3 | Electrocatalytic oxygen evolution performance of the NiFeB hydroxide nanosheets compared with the B-free NiFe hydroxide nanosheets. a** OER polarization curves (95% *iR* compensation). The current densities were normalized to the geometric area of the electrode. **b** Comparison of current densities observed with different catalysts at 1.53 vs. RHE. The current densities were normalized to the geometric area of the electrode and the electrochemically active surface area (ECSA), respectively. **c** Comparison of the OER activity of the NiFeB hydroxide nanosheets with those of the NiFe hydroxide nanosheets and typical nonnoble metal catalysts reported previously. Data were estimated from the OER polarization curves reported in refs. 13,22,34,54, and 20. **d** Differential pulse voltammetry curves (95% *iR* compensation). The currents for the NiFe hydroxide nanosheets were multiplied by 5 for clear demonstration (dotted curve). **e** Nyquist plots of the catalysts recorded at a constant potential of 1.50 V vs. RHE. Inset: equivalent circuit used in fitting the plot. **f** Chronopotentiometric curves for the catalysts at current densities of 10, 100, and 500 mA $cm_{geo}^{-2}$. No *iR* compensation was applied in the measurements.

catalysts prior to the OER, which was attributed to the $Ni^{2+}(OH)_2 \rightarrow Ni^{3+\delta}OOH$ transition. To avoid interference of the OER current densities by the Ni oxidation peak, we performed LSV of the catalysts with a cathodic sweep from 1.743 to 1.043 V vs. RHE (Fig. S8). The LSV curves were close to those obtained via anodic sweeps. The NiFeB hydroxide nanosheets reached a current density of 100 mA $cm_{geo}^{-2}$ at an overpotential of 256 mV, which was lower than that required by the B-free NiFe hydroxide nanosheets (overpotential, 329 mV). All of these

observations highlight the exceptional oxygen evolution activity of the NiFeB hydroxide nanosheets.

The $Ni^{2+}(OH)_2 \rightarrow Ni^{3+\delta}OOH$ transition was further investigated electrochemically by studying the Ni oxidation peak with differential pulse voltammetry (DPV) (Fig. 3d). The onset potentials of the Ni oxidation peaks were 1.33 and 1.39 V vs. RHE for the NiFeB and NiFe hydroxide nanosheets, respectively. Therefore, incorporation of B into the NiFe hydroxide catalyst caused a 60-mV cathodic shift of the

potential required to initiate oxidation of Ni. The peak position also underwent a cathodic shift of 20 mV with B. This result again highlighted the efficacy of B in promoting formation of high-oxidation-state Ni species at low overpotentials, consistent with the Raman and XAS observations. Notably, the Ni oxidation peak for the NiFeB hydroxide nanosheets was significantly larger than that of the NiFe hydroxide nanosheets, suggesting that a higher population of the active $Ni^{3+\delta}$OOH species was formed with this catalyst promoted by B. Both the lower formation potential and the increased population of the active $Ni^{3+\delta}$ species in the NiFeB hydroxide catalyst may have contributed to its superior OER activity.

To better evaluate the intrinsic activity of the NiFeB hydroxide nanosheets, we examined the current densities for the OER normalized to the ECSAs of the catalysts (Figs. 3b and S9). The ECSAs were calculated according to the double-layer capacitance of the nanosheets without supporting carbon (Fig. S10). The LSV data showed that the NiFeB hydroxide nanosheets outperformed the B-free NiFe hydroxide nanosheets. The current density of the NiFeB hydroxide nanosheets at 1.53 V vs. RHE was 31.8 mA $cm_{ECSA}^{-2}$, which was 11.0 times greater than that of the NiFe hydroxide nanosheets (2.9 mA $cm_{ECSA}^{-2}$). These results confirmed the critical role of B in improving the intrinsic catalytic activity of the NiFeB hydroxide catalyst for the OER.

Electrochemical impedance spectroscopy (EIS) was used to gain further insights into the kinetics of the OER with different catalysts. Figure 3e shows the Nyquist plots of the catalysts at 1.50 V vs. RHE. The plots could be fitted with a series connection of solution resistance ($R_0$) with two $R$-CPE units, each consisting of a charge transfer resistance ($R_1$ or $R_2$) and a constant phase element (CPE$_1$ or CPE$_2$, imperfect capacitors) arranged in parallel (equivalent circuit, Fig. 3e, inset; fitting parameters, Table S3). The two $R$-CPE units suggested two electron-transfer processes. One was the electrochemical oxidation of $H_2O$ with electron-transfer across the liquid−solid interface. The other was ascribed to electron transfer across the catalyst to the glassy carbon electrode. Apparently, the former was a more sluggish process than the latter. Therefore, of the two charge transfer resistances ($R_1$ and $R_2$), the larger one corresponded to the electrochemical charge transfer process. The electrochemical charge transfer resistances of the NiFe and NiFeB hydroxide nanosheets were therefore determined to be 21.0 and 6.2 Ω, respectively. The NiFeB hydroxide catalyst showed a much lower charge transfer resistance than its NiFe hydroxide counterpart, which agreed well with its exceptional OER activity.

The single-layer NiFeB hydroxide nanosheets also showed high stability in the OER. Figure 3f shows chronopotentiometric curves of the NiFeB and NiFe hydroxide nanosheets at constant current densities of 10, 100, and 500 mA $cm^{-2}$ without $iR$ compensation. The potential required by the NiFeB hydroxide nanosheets to maintain the current density was kept stable for 130 h. TEM and ICP−MS analyses confirmed that after electrocatalysis, the ultrathin NiFeB hydroxide nanosheets essentially retained their initial morphology and composition (Figs. S11, S12, and Table S4), which contributed to the stability of the catalyst in the alkaline OER.

## DFT calculations

We carried out DFT calculations to gain atomic-scale insights to interpret the role of B in improving the OER activity of the NiFeB hydroxide nanosheets (Fig. 4). The simulation model for the active species of the NiFe hydroxide catalyst was constructed as (Ni, Fe)OOH, which was adopted from the research by Goddard et al.[55]. Three of the 12 Ni atoms were replaced with Fe atoms. Our work involves doping of B into the above NiFe-based catalyst. According to the experiments, three B atoms were incorporated into the lattice at the tetrahedral centers of four oxygen atoms. After geometry optimization, each B formed three bonds with strong valence characters with three neighboring oxygen atoms. As the exact B positions could not be

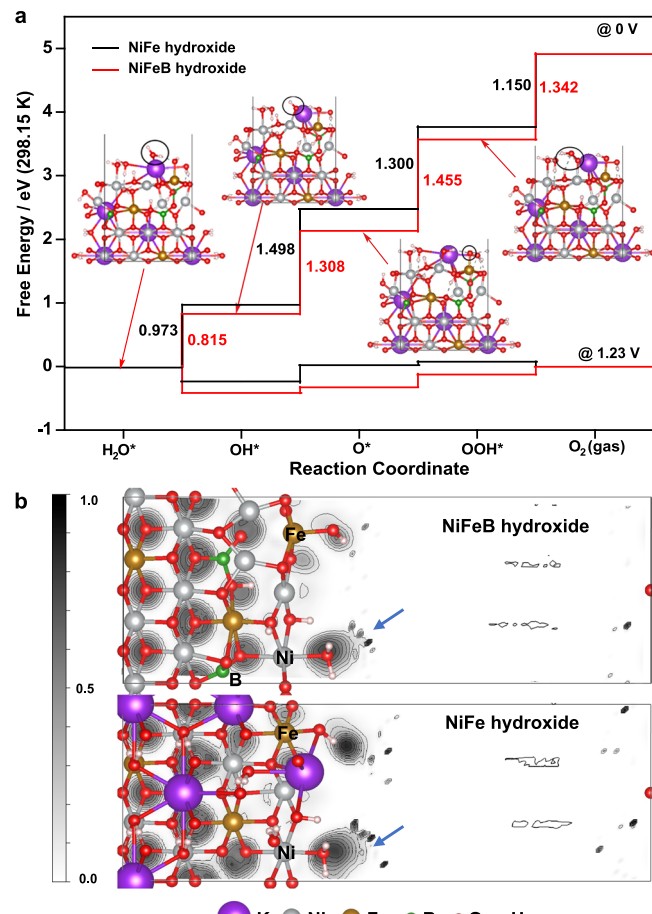

**Fig. 4 | DFT calculation results. a** Free energy diagram for the OER and the reactive intermediates involved with two different catalysts, i.e., NiFeB hydroxide and NiFe hydroxide. Inset: DFT optimized geometries of the intermediates on the NiFeB hydroxide catalyst. **b** Electron localization functions (ELFs) of the NiFeB (upper) and NiFe (lower) hydroxide catalysts from DFT calculations. The scale bar is on the left with a gray-scale scheme for viewing convenience. The silver, yellow, red, white, green, and purple atoms represent Ni, Fe, O, H, B, and K (counter ions), respectively.

determined experimentally, a random search algorithm was used to explore possible doping structures, and the most energetically favorable structure after DFT optimization was employed to simulate the OER reactions. As we used KOH as the electrolyte for the OER, $K^+$ was introduced as the cation to balance the negatively charged framework. The DFT-optimized structure is shown in Fig. S13.

Introduction of B decreased the energy barrier of the OER, as indicated by the DFT calculations. The free energies of OER intermediates were investigated to predict the OER activities of the NiFeB hydroxide and NiFe hydroxide catalysts (Fig. 4a). We considered a four-step pathway with *OH, *O, and *OOH intermediates on the reaction pathway from *$H_2O$ to $O_2$. The free energies were calculated at two potentials (0 and 1.23 V), which showed similar trends for the free energy changes along the reaction coordinates. At 0 V, *O formation (*OH → *O) with the NiFe hydroxide was the potential-determining step (PDS) with a free energy difference of 1.498 eV, consistent with previous results[55]. After introducing B, the formation energy of *O on the NiFeB hydroxide decreased to 1.308 eV. Instead, the formation energy of *OOH (*O → *OOH) increased to 1.455 eV. As a result, *OOH formation became the PDS. Therefore, the PDS underwent a forward shift with a decrease in the free energy barrier (NiFe hydroxide, 1.498 eV; NiFeB hydroxide, 1.455 eV), which explained the improved OER activity

and indicated the critical role of B in enhancing the OER activity of the NiFeB hydroxide catalyst.

DFT calculations also suggested an elevated oxidation state of Ni after introducing B into the NiFe hydroxide. These electronic changes were indicated by the ELFs (Fig. 4b). A gray-scale scheme was employed to elucidate the ELFs, in which 0.0 (white) stands for a fully delocalized state and 1.0 (black) stands for a fully localized state. Electron localization by the Ni-bound OH (indicated by the arrows in the image) was enhanced after introducing B, which provided indirect evidence for the increased oxidation states of Ni in proximity to B. To quantitatively evaluate these charges, we carried out a Bader charge analysis, which showed a maximum of +0.06 in the formal charge. The electronic interaction made it possible for B to impose an electron sink effect on the Ni species and promote the oxidation state transition, leading to an effectively reduced onset potential for the OER and therefore excellent catalytic activity.

The roles of Ni and Fe in the NiFeB hydroxide were further investigated with a density of states (DOS) analysis and projected DOS (pDOS) analysis (Fig. S14). It appeared that the electronic states near the Fermi level were predominantly from the Ni and Fe 3$d$ orbitals. This analysis indicated that both Ni and Fe serve as catalytic active sites by contributing the states near the valance band edge. Indeed, such a conclusion is consistent with our proposed mechanism, in which Fe provides the adsorption sites for *O and Ni provides the adsorption sites for other OER intermediates (Fig. 4a, inset).

## Discussion

In summary, we have successfully synthesized single-layer NiFeB hydroxide nanosheets via facile hydrolysis of NiFeB alloy nanoparticles in an alkaline medium. TEM, XRD, and XPS results confirmed formation of a single $MO_6$ layer (thickness, ~0.5 nm) without stacking of the layers. The NiFeB hydroxide nanosheets exhibited superior catalytic activity and remarkable long-term stability for the OER in an alkaline electrolyte. An overpotential of only 252 mV was required to reach a current density of 100 mA $cm_{geo}^{-2}$, which outperformed most Earth-abundant and nonnoble metal catalysts reported in the literature for the OER. Raman spectroscopy, XAS, and electrochemical DPV analyses proved that the OER activity was associated with the transition of $Ni^{2+}$ to $Ni^{3+\delta}$, the potential of which was effectively reduced by incorporating B into the NiFe hydroxide catalyst. More specifically, DPV analysis showed that incorporation of B into the NiFe hydroxide catalyst has caused a 60-mV cathodic shift of the potential required to initiate the oxidation state transition of Ni. DFT results indicated that B increased the oxidation state of Ni, shifted the potential-determining step of the OER, and reduced the energy barrier, which explains the experimental observations. All of these results supported our hypothesis for the critical role of B in promoting the formation of high-oxidation-state $Ni^{3+\delta}$ as the active species boosting the OER kinetics. Overall, our findings included the following: (1) Incorporating B into NiFe hydroxide proved to be an effective route to constructing highly efficient catalysts for the OER; (2) Incorporation of B in NiFeB hydroxide reduced the potential required for oxidation of $Ni^{2+}$ into the active $Ni^{3+\delta}$ species, which contributed to the decreased onset potential and enhanced kinetics of the OER. (3) Electron-deficient B may have served as an electron sink to promote the oxidation of $Ni^{2+}$ to $Ni^{3+\delta}$ during the electrocatalytic OER. We believe our study provides a robust strategy for electronic engineering of Ni-based catalysts for the efficient OER in high-performance electrochemical water splitting applications.

## Methods

### Materials

Nickel(II) nitrate hexahydrate [$Ni(NO_3)_2 \cdot 6H_2O$] and iron(III) nitrate nonahydrate [$Fe(NO_3)_3 \cdot 6H_2O$] were purchased from Sinopharm. Sodium borohydride ($NaBH_4$, 99%) was purchased from Acros Organics. Sodium hydroxide (AR 96%) was purchased from Aladdin. All chemicals were used as received without further purification.

### Synthesis of single-layer NiFeB hydroxide nanosheets

In a typical synthesis, 727 μL of 0.14 M $Ni(NO_3)_2$ and 673 μL of 0.05 M $Fe(NO_3)_3$ were dissolved in 10 mL of ethanol. After sonication for 5 min, 1 mL of 0.5 M $NaBH_4$ was added dropwise into the above solution. The mixed solution was sonicated for another 45 min. The solid product (NiFeB alloy nanoparticles) was collected by centrifugation and washed with ethanol. Then, an equivalent volume of KOH (1 M in $H_2O$) was added to the above ethanolic solution to transform the NiFeB nanoparticles into single-layer NiFeB hydroxide nanosheets.

### Synthesis of B-free NiFe hydroxide nanosheets

B-free NiFe hydroxide nanosheets were synthesized by following the typical protocol for the synthesis of NiFeB hydroxide nanosheets, except that 1 mL of 0.5 M NaOH was used in place of 1 mL of 0.5 M $NaBH_4$.

### Characterizations

TEM was performed on a Hitachi HT-7700 microscope operated at 100 kV. HRTEM and EDS were performed on a JEOL JEM-F200 microscope operated at 200 kV. XRD patterns were recorded on a Rigaku SmartLab Powder X-ray diffractometer equipped with Cu Kα radiation and a D/TEX Ultra detector. ICP–MS was conducted on a NexION 350D system. XPS data were collected on a Thermo Fisher ESCALAB Xi+ spectrometer equipped with monochromatic Al Kα radiation. All binding energies in XPS studies were corrected by referring to the C 1$s$ binding energy at 284.5 eV. Raman spectra were collected on a Horiba Jobin Yvon LabRAM HR800 system equipped with a 532-nm laser. XAS measurements were carried out in total electron yield mode under an ultrahigh vacuum at beamline 4B9B of the Beijing Synchrotron Radiation Facility at the Institute of High Energy Physics, Chinese Academy of Sciences. Au 4$f_{7/2}$ core-level spectra were recorded for photon energy calibration.

### Electrochemical measurements

Electrochemical measurements were performed with a CHI 760E electrochemical workstation (CH Instruments, Inc.) equipped with a Pine rotating disk system unless stated otherwise at room temperature (25 °C). A three-electrode configuration was employed, including a working electrode (RDE, 0.196 $cm^2$), a reference electrode (saturated calomel electrode, SCE), and a counter electrode (graphite rod). Before use, the SCE electrode was calibrated by measuring the reversible hydrogen electrode potential using a Pt electrode under $H_2$ atmosphere. All potentials were converted to values relative to the RHE. The OER overpotentials were calculated by subtracting 1.23 V from the potentials vs. RHE. The metal hydroxide catalyst was mixed with Vulcan XC72R carbon black (loadings, NiFeB hydroxide: 91.6 wt%; NiFe hydroxide: 93.5 wt%) and dispersed in a mixture of ethanol/$H_2O$/Nafion (40 wt%) = 3:1:0.3 (volume ratio) to form a homogeneous ink. An aliquot of the ink was dried on the RDE with metal hydroxide loadings of 0.94 mg $cm_{geo}^{-2}$ (NiFeB hydroxide) and 1.25 mg $cm_{geo}^{-2}$ (NiFe hydroxide) (Table S1). OER polarization curves were obtained using LSV in 1 M KOH at a scan rate of 10 mV $s^{-1}$ with 95% $iR$ compensation ($iR$, automatically measured with the CHI 760E electrochemical workstation). All LSV polarization curves were recorded after activating the catalysts with CV (1.043–1.743 V vs. RHE) until a stable state was reached. DPV was performed from 1.043 V to 2.043 V vs. RHE with the following experimental parameters: pulse amplitude, 0.05 V; pulse width, 0.05 s; pulse period, 0.5 s; 95% $iR$ compensation. The EIS was obtained with an Autolab PGSTAT302N electrochemical workstation at 1.5 V vs. RHE over the frequency range from 1 MHz to 0.1 Hz in 1 M KOH. ECSAs were determined by measuring the capacitive current associated with double-layer charging from the scan rate dependence

of CV. The potential window was 0.943–0.993 V vs. RHE. The double-layer capacitance, $C_{dl}$, was estimated from the slope obtained from a plot of the current density at 0.968 V versus the scan rate. The ECSA was calculated to be $C_{dl}/(C_s\,m)$, where $C_s$ was the specific capacitance with a value of 0.04 mF cm$^{-2}$ [34,56,57] and $m$ was the loading of catalyst on the electrode (units: g m$^{-2}$). The stabilities of the catalysts were measured by chronopotentiometry. The catalysts were supported on 1 cm × 1 cm carbon cloths with a loading of 0.4 mg cm$^{-2}$. The potentials required for maintaining current densities of 10, 100, and 500 mA cm$^{-2}$ were recorded for 90–130 h.

## DFT calculations

The structure of the NiFeB hydroxide catalyst was simulated by adapting the model in the literature by Goddard et al.[55] Three of the 12 Ni atoms were substituted by Fe, after which 3 B atoms were incorporated into the lattice. A vacuum area of 15 Å was placed to avoid interactions of the slabs. The top two atomic layers were relaxed during the geometry optimizations, while the bottom two layers were kept fixed as in the bulk structure. DFT calculations were performed by using the Vienna Ab initio Simulation Package (VASP), version of 5.4.4[58–60], with the projector augmented wave (PAW) method[61] and a plane wave basis set. The DFT calculations used the generalized gradient approximations (GGA) of the Perdew–Burke–Ernzerhof (PBE) functional[62]. A dispersion correction, from the DFT-D3 method with Becke–Jonson damping was included in the calculations[63]. The energy cutoff was set to 400 eV. The Γ centered (3 × 3 × 1) Monkhorst−Pack k-point grids were used for slab calculations. Spin polarization was included in the calculations. The self-consistent electronic steps were considered converged when the changes in both total energy and eigenvalues between the two steps were smaller than $1 \times 10^{-5}$ eV. The partial occupancies for each orbital were set with the first-order Methfessel-Paxton scheme with the smearing width of 0.01 eV. The dipole moment corrections for the total energy were applied in the direction normal to the surface. The vibrational frequencies were computed to consider the zero-point energies, enthalpy, and entropy and ultimately calculate the free energies at room temperature (298.15 K). To understand the origin of the increased OER activity in NiFeB hydroxide, we carried out a DFT calculation considering all possible intermediates and elementary reactions of the OER. Typically, a four-step mechanism was employed to investigate the reaction kinetics, and formation of the *O intermediate was considered the potential-determining step. We employed such a reaction pathway in our calculations.

## Data availability

The data supporting the findings of this study are available within the paper and the Supplementary Information. Source data are provided with this paper.

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

## Acknowledgements

C.G. acknowledges the support of the National Natural Science Foundation of China (22071191), the Key Research and Development Projects of Shaanxi Province (2021GXLH-Z-022), the Key Scientific and Technological Innovation Team of Shaanxi Province (2020TD-001), the Fundamental Research Funds for the Central Universities, and the World-Class Universities (Disciplines) and the Characteristic Development Guidance Funds for the Central Universities. T.C. was supported by Suzhou Key

Laboratory of Functional Nano & Soft Materials, Collaborative Innovation Center of Suzhou Nano Science & Technology, the Priority Academic Program Development of Jiangsu Higher Education Institutions (PAPD), the 111 Project, the National Natural Science Foundation of China (21903058), and the Natural Science Foundation of Jiangsu Province (BK20190810). Y.Y appreciates the support from the National Natural Science Foundation of China (22172153), Users with Excellence Program of Hefei Science Center CAS (2021HSC-UE001), and funding support from the University of Science and Technology of China (USTC). The authors thank the Instrumental Analysis Center of Xi'an Jiaotong University for assistance with the HRTEM and XPS measurements.

## Author contributions

C.G. conceived the idea and supervised the project. Y.B. carried out the syntheses, characterizations, and electrocatalytic experiments. Y.W. and M.X. carried out the DFT calculations. T.C. supervised the DFT calculations. Y.Y., K.N., and J.W. contributed to the XAS analyses. X.Z, Z.Z., and Z.L. contributed to the materials characterizations and electrocatalytic measurements. C.G., Y.B., and T.C. wrote the manuscript. All authors discussed the results and revised the manuscript.

## Competing interests

The authors declare no competing interests.
