## [Peer Review File · Nature Communications]

Promoting Nickel Oxidation State Transitions in Single-Layer NiFeB Hydroxide Nanosheets for Efficient Oxygen EvolutionREVIEWER COMMENTS

Reviewer #1 (Remarks to the Author):

This is an interesting work that provides useful and novel guidelines for improving the performances of Oxygen evolution reaction. The authors identify a clear thesis, i.e. the beneficial effect of B in promoting the formation of high-oxidation Ni species, and try to prove it by combining different experimental and theoretical approaches.

The manuscript is well written, the methods are described with sufficient detail, the results are presented clearly and well supported by high-quality figures. Overall the results indeed points into the direction of the thesis.

In my opinion there are some issues that could be improved so as to better support the thesis. Detecting and explaining the change of Ni oxidation states would be better achieved with more specific spectroscopies that could complement the Raman measurements, whose interpretation in this case is not that straightforward (Ni²⁺ vs Ni^{3+/4+}). I am particularly concerned by the DFT calculations:

- * It is not clear how close and how related is the model system employed in the situations to the actual catalyst. A better connection between the structures of the model system and of the catalyst is necessary.

- * The charge analysis ELF is not that illustrative of the electronic effects on Ni driven by B atoms. The plots shown in fig. 4 b depend on the cutting plane chosen to analyze the data. From this point of view an integrated measure would be much more significative.

I would be happy to review again the manuscript after these issues have been addressed.

Reviewer #2 (Remarks to the Author):

This part of work prepared an electrode of NiFeB hydroxide nanosheets, which displayed a low overpotential and high activity for electrocatalytic water oxidation in alkaline solution. Raman spectroscopic and electrochemical studies demonstrated the role of doped-B in promoting the formation of high-oxidation-state Ni, and DFT calculations suggested the decrease of the energy barrier for the rate-determining step of OER by doping B to the NiFe hydroxide. These results could be interesting for researchers engaging in hydrogen production from water splitting. The manuscript might be suitable for publication in Nat. Commun. after the following problems are addressed.

(1) In the manuscript, the authors used RuO₂ as one of the reference catalyst. It is not proper to compare the electrocatalytic activity of RuO₂ with that of their NiFeB catalyst in alkaline media, because RuO₂ is a benchmark OER catalyst in acidic media instead of basic media. No earth-abundant metal-based OER catalysts can outperform the RuO₂ in acidic media to date. Comparison of the performance of a catalyst in its favored media with the performances of other catalysts in their disfavored media is meaningless.

(2) Page 4: "We hypothesize that B as an electron-deficient element should withdraw electrons from Ni, promoting the formation of the desired Ni^{3+/4+} species at low overpotentials, which is beneficial to the OER." In fact, when the electron-deficient B withdraws electrons from Ni in the as-prepared NiFeB alloy, the Ni proximate to the B element in NiFeB alloy would have less electron-density. In this case, it should be more difficult to be oxidized, and the overpotential should be shifted to positive direction. Therefore, simply attributed the promoting formation of Ni^{3+/4+} species to the electron-withdrawing effect of B seems problematic. Some other rational reasons for the promoted OER activity of NiFeB should be clearly explained.

(3) According to Figure 2, in addition to Ni, the Fe in NiFeB catalyst is also much easier to be oxidized compared to that in B-free NiFe catalyst. What is the state of the oxidized Fe in NiFeB hydroxide catalyst? Does the different oxidation states of Fe in NiFeB and NiFe hydroxides have an influence on the electrocatalytic performances of these OER catalysts?

(4) The authors repeatedly use the word “confirm” in the discussion of DFT calculations. The results of DFT calculations cannot confirm the role of B in the NiFeB hydroxide nanosheets for OER. The following sentences should be revised.

Abstract: “DFT calculations confirm an elevated oxidation state of Ni conjugated to B”. It is better to change it to “DFT calculations suggest”.

Page 14: Similarly, the sentence “..... confirms the critical role of B in enhancing the OER activity” is better revised to “indicates the critical role of B”.

Page 15: The sentence “DFT calculations also confirm” is better changed to “DFT calculations also suggest”.

Page 15: The sentence “These results confirm that introducing B into” is better revised to “These results show that introducing B into”.

Page 15: Change the sentence “DFT results confirm that B increases the oxidation state of Ni” to “DFT results indicate that B increases”.

(5) The method employed for the iR-compensation of LSVs should be clearly described. Was it derived from electrochemical workstation (automatically) or EIS Nyquist plots (manually)?

(6) In Nyquist plots (Figure 3e), the scales for the x- and y-axis should be the same or similar to make the curve shape as a semicircle.

(7) Page 9, the last line: “loaded on glass carbon electrodes” should be “loaded on glassy carbon electrodes”.

(8) In this work, the long-term chronopotentiometric experiments were carried out at a current density of 10 mA cm⁻². To demonstrate that the catalyst is promising for application in electrocatalytic water splitting, the results of long-term chronopotentiometric experiments at 100 and even 500 mA cm⁻² as well as the post-analysis of the used electrodes and electrolytes should be provided.

(9) Why when the electrons of Ni are withdrawn by the electron-deficient B proximate to it, the ELF of Ni is enhanced?

(10) It is better to give the loading amounts of Ni, Fe, and B on the electrodes in Table S1.

(11) The very related work on the CoFeB OER catalyst published very recently in Journal of Colloid and Interface Science (DOI:10.1016/j.jcis.2021.07.024) should be cited.

Reviewer #3 (Remarks to the Author):

Bai et al. present a synthesis of single-layer NiFeB hydroxide nanosheets and demonstrate the efficacy of electron-deficient boron in promoting the formation of high-oxidation-state Ni for improved OER activity. They reported that a current density of 100 mA cm⁻² has been achieved in 1.0 M KOH at an overpotential of 252 mV, placing it among the best Ni-based catalysts for the OER. After carefully reading, it looks more like a pretty typical high surface area layered (oxy)hydroxide paper. It's hard to judge how much of the improvement is from fundamental changes in electronic structure vs differences in surface area and morphology. Not that high current isn't important for devices, just more engineering. Moreover, a recent paper (<https://www.nature.com/articles/s41467-021-26307-7>)

about NiFe-Borate catalyst just came out. Therefore, the novelty of this study is not high as required by Nature Communications and I don't recommend publishing this paper.

Several specific comments are shown below:

1. RuO₂ is not a benchmark reference material in 1M KOH.
2. In Figure S3, why do Ni 2p and Fe 2p spectra of NiFeB alloy show much noise? While both look much better for NiFeB hydroxide.
3. The authors should explain why Ni redox peaks for NiFe hydroxide and NiFeB hydroxide are different. Are they directly related to the Ni valence state change? And how do these Ni redox peaks change when changing the scan rate during OER measurements?
4. Why does introducing B into NiFe only affect the Ni oxidation state? Is there any chance it will also affect the Fe oxidation state?
5. In Figure 4b, why does a higher ELF of 1.22 denote a higher Ni oxidation state?

Response to Reviewers

Dear reviewers,

Thank you very much for offering us the valuable comments and suggestions, which have been very helpful for us to improve our manuscript. We here respond point by point and submit a revised manuscript with all changes highlighted. We hope you would find that all your concerns have been appropriately addressed.

To make it easy to read, we have marked all our responses in blue.

Reviewer #1: This is an interesting work that provides useful and novel guidelines for improving the performances of oxygen evolution reaction. The authors identify a clear thesis, i.e. the beneficial effect of B in promoting the formation of high-oxidation Ni species, and try to prove it by combining different experimental and theoretical approaches. The manuscript is well written, the methods are described with sufficient detail, the results are presented clearly and well supported by high-quality figures. Overall the results indeed points into the direction of the thesis.

Response: We thank the reviewer for his/her kind recognition of the value of this work and the valuable comments to us. We here respond point by point.

In my opinion there are some issues that could be improved so as to better support the thesis. Detecting and explaining the change of Ni oxidation states would be better achieved with more specific spectroscopies that could complement the Raman measurements, whose interpretation in this case is not that straightforward (Ni^{2+} vs $Ni^{3+/4+}$). I am particularly concerned by the DFT calculations:

Response: We are grateful to the reviewer for the kind suggestion. In our previous submission, we have examined the oxidation states of Ni at different overpotentials by Raman spectroscopy. This technique has been widely used to reveal the oxidation state of Ni in OER catalysts, as it detects characteristic vibration/rotation modes of the chemical bonds in different reaction intermediates. We agree with the reviewer that the study of the oxidation states of Ni could be supplemented by other spectroscopies. Therefore, we carried out X-ray absorption spectroscopy (XAS) to gain further insights into the oxidation state transition of Ni in the catalysts. The results agree with the Raman spectroscopy results, thus strengthening our understanding of the B-promoted oxidation state transition of Ni in the NiFeB hydroxide catalyst.

Besides, we also supplemented some results on the oxidation state transition of Fe in both the NiFeB and NiFe hydroxide catalysts by the XAS. The results show that the oxidation state of Fe does not change significantly with increasing overpotentials. Therefore, the active sites of the OER can be confirmed to be $Ni^{3+\delta}OOH$ rather than the Fe species. Nevertheless, the incorporation of Fe into the catalysts has been proven to have contributed to the boosted oxidation state transition of Ni in the

hydroxide catalyst.

The XAS results have been added to Figure 2c, d of the revised Manuscript and Figure S6 of the revised Supporting Information. The detailed discussion has been added to pages 8–11 of the revised Manuscript.

1. It is not clear how close and how related is the model system employed in the situations to the actual catalyst. A better connection between the structures of the model system and of the catalyst is necessary.

Response: We agree with the reviewer that we should have provided more details of the connection between the simulation models and the experiment.

The structure of the NiFeB hydroxide nanosheets has been examined experimentally by XRD, HRTEM, XPS, and XAS to be a single layer of edge-sharing MO_6 ($M = Ni, Fe$) octahedra with B incorporated into the single layer via B–O–M ($M: Ni, Fe$) bridging bonds. The active species for the OER is NiOOH formed at high overpotentials. We construct a model for the DFT calculations to reflect this structure.

The simulation model for the active species of the NiFe hydroxide catalyst is constructed as Fe-doped NiOOH, i.e., (Ni, Fe)OOH, adopted from the research by Goddard et al. (ref. 55). Such a model is close to that reported by Friebel et al. (*J. Am. Chem. Soc.* 2015, 137, 3, 1305) but with explicit cations, which can well reproduce the overpotential as observed experimentally. Our work is to dop B into the above NiFe-based catalyst. According to the experiment, B is inserted into the tetrahedron center of four oxygen atoms, keeping the same molar concentrations as the experiment. After geometry optimization, B forms three bonds with strong valence characters with three neighboring oxygen atoms. As the experiment cannot determine the exact B positions, a random searching algorithm is used to explore possible doping structures, and the most energetically favorable structure after DFT optimization is employed to simulate the OER reactions. As we use KOH as the electrolyte for the OER, K^+ is introduced as the cations to balance the negatively charged framework. A monolayer structure is desirable, according to the experiment. Indeed, the simulation includes just one monolayer of the framework, corresponding to the experiment (single-layer structure), where the periodic boundary condition has little effect on the reactions. Therefore, our theoretical model closely matches the experiment within the computation cost.

The details of the theoretical modeling have now been included on pages 14–15 of the revised Manuscript.

2. The charge analysis ELF is not that illustrative of the electronic effects on Ni driven by B atoms. The plots shown in fig. 4 b depend on the cutting plane chosen to analyze the data. From this point of view an integrated measure would be much more significant. I would be happy to review again the manuscript after these issues have been addressed.

Response: We agree with the reviewer that the electron localization function (ELF) is useful for

quantifying geographical localization but not for quantifying the oxidation state caused by B withdrawing electrons from Ni. Therefore, we include the Bader charge analysis that approximates the total electronic charge of an atom by integrating the electron density within Bader volume, which can be used to quantify the cost of removing the charge from an atom. We add the following discussion on page 16 of the revised Manuscript.

“To evaluate the charges quantitatively, we carried out the Bader charge analysis, which shows a maximum of +0.06 in the formal charge of Ni due to the electron withdrawing by B.”

Reviewer #2: *This part of work prepared an electrode of NiFeB hydroxide nanosheets, which displayed a low overpotential and high activity for electrocatalytic water oxidation in alkaline solution. Raman spectroscopic and electrochemical studies demonstrated the role of doped-B in promoting the formation of high-oxidation-state Ni, and DFT calculations suggested the decrease of the energy barrier for the rate-determining step of OER by doping B to the NiFe hydroxide. These results could be interesting for researchers engaging in hydrogen production from water splitting. The manuscript might be suitable for publication in Nat. Commun. after the following problems are addressed.*

Response: We thank the reviewer for his/her kind recognition of the value of this work and the valuable comments to us. We here respond point by point.

(1) *In the manuscript, the authors used RuO₂ as one of the reference catalyst. It is not proper to compare the electrocatalytic activity of RuO₂ with that of their NiFeB catalyst in alkaline media, because RuO₂ is a benchmark OER catalyst in acidic media instead of basic media. No earth-abundant metal-based OER catalysts can outperform the RuO₂ in acidic media to date. Comparison of the performance of a catalyst in its favored media with the performances of other catalysts in their disfavored media is meaningless.*

Response: Thanks for pointing out the inappropriate reference catalyst in our study. Our initial purpose of using RuO₂ as a reference was to make the performance of our catalyst comparable with literature-reported catalysts, because the OER activity of the commercial RuO₂ catalyst has been widely reported in the literature as a reference catalyst even in alkaline media. We agree with the reviewer that RuO₂ is a benchmark OER catalyst in acidic media, not in alkaline media. Transition metal oxides or hydroxides, NiFe layered double hydroxide (NiFe-LDH) for example, show much higher OER activity than the commercial RuO₂. To avoid the confusion caused to the reviewer and other readers, we decide to remove the OER data of RuO₂. Instead, we compare the OER activity of the NiFeB hydroxide nanosheets with the B-free counterpart, which already shows the advantage of B-doping in improving the OER activity of the hydroxide catalyst.

(2) *Page 4: “We hypothesize that B as an electron-deficient element should withdraw electrons from Ni, promoting the formation of the desired Ni^{3+/4+} species at low overpotentials, which is beneficial to the OER.” In fact, when the electron-deficient B withdraws electrons from Ni in the as-prepared NiFeB alloy, the Ni proximate to the B element in NiFeB alloy would have less electron-density. In this*

case, it should be more difficult to be oxidized, and the overpotential should be shifted to positive direction. Therefore, simply attributed the promoting formation of $\text{Ni}^{3+/4+}$ species to the electron withdrawing effect of B seems problematic. Some other rational reasons for the promoted OER activity of NiFeB should be clearly explained.

Response: We thank the reviewer for the comments. We agree with the reviewer that the electron density of Ni may affect its oxidation state transition. The electronic interaction is complicated and may show different characteristics according to the catalytic systems. For the NiFeB hydroxide catalyst we investigated, the Raman spectroscopy and X-ray absorption spectroscopy (XAS, new results during the revision) results clearly show the critical role of B in promoting the oxidation state transition of Ni from +2 to +3 and higher, consistent with the exceptional catalytic activity in the OER. B is unique from many other common elements because it is intrinsically electron-deficient, which may withdraw electrons from adjacent Ni atoms to accelerate their oxidation state transition under applied potentials. The strong efficacy of this unique element, irreplaceable by other elements, points to its intrinsic electron-withdrawing effect. To verify it, we carried out DFT calculations to reveal the atomic electronic interactions in the NiFeB hydroxide catalyst. The results show that the charge density (an indicator of the oxidation state) of Ni in the presence of B is significantly elevated, suggesting that the electron-withdrawing property of B is responsible for the promoted oxidation state transition of Ni in the catalyst. The DFT calculation results well support our hypothesis.

The strong effect of the electron transfer on the promoted OER activity is consistent with some previous results in the literature. Examples include: (1) Boettcher et al. (*J. Am. Chem. Soc.* 2014, 136, 6744) attributed the improved OER activity of the Ni-Fe oxyhydroxide catalyst to the electron-withdrawing effect of Fe on Ni. To consolidate it, they investigated the OER activities of NiOOH of different layers on an electronegative Au substrate. With decreasing layer numbers, the OER activity increased by more than 2 orders of magnitude, suggesting the strong effect of the $\text{NiOOH} \rightarrow \text{Au}$ electron transfer on the OER activity. (2) Ye et al. (*J. Am. Chem. Soc.* 2016, 138, 9128) found that a Ni-based catalyst showed improved OER activity when conjugated with noble metal nanoparticles. Under visible light irradiation, positively charged “holes” are produced in noble metal nanoparticles due to the surface plasmon resonance, which readily increases the oxidation state of Ni in their proximity, leading to promoted OER activity. (3) Sargent et al. (*Nat. Chem.* 2018, 10, 149) took the view that generating high-oxidation-state transition-metal sites at low overpotentials should improve the OER activity. They synthesized NiCoFeP oxyhydroxides and proved by in situ soft XAS that P promotes the transition metals to high oxidation states (Ni^{4+}) at low overpotentials studies, which accounts for the improved OER activity. All these prior works suggest that the electron-withdrawing effects may play a dominant role in promoting the oxidation state transition of transition metals for improved OER activity.

Again, we thank the reviewer for the insightful comments. All the current data we collected strongly support the role of the electron-withdrawing property of B in boosting the OER activity, which may guide the design of highly efficient catalysts for this important reaction. We would also like to dig deeper into the system to gain more insights into the detailed electronic interaction mechanisms in our

future study.

(3) According to Figure 2, in addition to Ni, the Fe in NiFeB catalyst is also much easier to be oxidized compared to that in B-free NiFe catalyst. What is the state of the oxidized Fe in NiFeB hydroxide catalyst? Does the different oxidation states of Fe in NiFeB and NiFe hydroxides have an influence on the electrocatalytic performances of these OER catalysts?

Response: We thank the reviewer for the comments. The Raman signal at $\sim 680\text{ cm}^{-1}$ may originate from Fe species such as Fe_3O_4 , $\gamma\text{-Fe}_2\text{O}_3$, and $\gamma\text{-FeOOH}$. However, it is difficult to discriminate the oxidation state of Fe by Raman spectroscopy (Bell et al., *J. Am. Chem. Soc.* 2013, 135, 12329). Therefore, we further examined the evolution of the oxidation state of Fe in the NiFeB and B-free NiFe hydroxide nanosheets at different overpotentials by Fe L_{3,2}-edge X-ray absorption spectroscopy (XAS) (Figure S6). For both the catalysts, the spectra show an intense peak ii (Fe 2p \rightarrow 3d e_g transition) relative to peak i (Fe 2p \rightarrow 3d t_{2g} transition) with an intensity ratio of ~ 2.4 , corresponding to Fe³⁺ in the catalysts. The spectra display no significant changes with increasing overpotentials, suggesting that the oxidation state of Fe is virtually unchanged during the catalysis. Therefore, our conclusion is that the oxidation state of the Fe is +3 in both NiFeB and NiFe hydroxide catalysts as synthesized and during the electrocatalysis.

It is worth noting that Fe species did contribute to the boosted oxidation state transition of Ni in the hydroxide nanosheets. Raman spectroscopy suggests that the overpotential required for the Ni²⁺(OH)₂ \rightarrow Ni^{3+ δ} OOH transition increased to 166 mV in a Fe-free NiB hydroxide catalyst (Figure S5), which is significantly higher than the overpotential required by a NiFeB hydroxide catalyst observed by the same technique (66 mV, Figure 2a). Therefore, Fe is indispensable in the hydroxide catalysts toward high catalytic efficiencies.

The results have been added to Figure S5, S6 of the revised Supporting Information and discussed on pages 8–9, 10–11 of the revised Manuscript.

(4) The authors repeatedly use the word “confirm” in the discussion of DFT calculations. The results of DFT calculations cannot confirm the role of B in the NiFeB hydroxide nanosheets for OER. The following sentences should be revised.

Abstract: “DFT calculations confirm an elevated oxidation state of Ni conjugated to B”. It is better to change it to “DFT calculations suggest”.

Page 14: Similarly, the sentence “..... confirms the critical role of B in enhancing the OER activity” is better revised to “indicates the critical role of B”.

Page 15: The sentence “DFT calculations also confirm” is better changed to “DFT calculations also suggest”.

Page 15: The sentence “These results confirm that introducing B into” is better revised to “These results show that introducing B into”.

Page 15: Change the sentence “DFT results confirm that B increases the oxidation state of Ni” to “DFT results indicate that B increases”.

Response: We thank the reviewer for the kind suggestions. The corrections have been made to the revised Manuscript.

(5) *The method employed for the iR -compensation of LSVs should be clearly described. Was it derived from electrochemical workstation (automatically) or EIS Nyquist plots (manually)?*

Response: We have added experimental details of the iR compensation to the revised “Materials and Methods” section (page 19). The iR compensation was made directly by the CHI 760E electrochemical workstation, with the iR values probed automatically by the electrochemical workstation.

(6) *In Nyquist plots (Figure 3e), the scales for the x- and y-axis should be the same or similar to make the curve shape as a semicircle.*

Response: According to the reviewer’s suggestion, we have adjusted the scales for the x- and y-axis of the Nyquist plots. We would like to explain that the Nyquist plots are not perfect semicircles but flatted ones to some extent. Similar results are found in the literature, such as *Nat. Commun.* 2016, 7, 11981; *Adv. Mater.* 2019, 31, 1900178; and *ACS Catal.* 2019, 9, 2, 1013. This is because the plots cannot be well fitted by a series of RC circuits. They can be well fitted by replacing the capacitance unit (C) with a constant phase element (CPE, an imperfect capacitor, i.e., a capacitor with a non-90° $E-I$ phase angle). Compared with a perfect capacitor, CPE is particularly suited for fitting the Nyquist plots obtained from a practical system that deviates from an ideal one. The detailed fitting parameters are presented in Table S3 of the revised Supporting Information.

(7) *Page 9, the last line: “loaded on glass carbon electrodes” should be “loaded on glassy carbon electrodes”.*

Response: We have corrected this typo.

(8) *In this work, the long-term chronopotentiometric experiments were carried out at a current density of 10 mA cm^{-2} . To demonstrate that the catalyst is promising for application in electrocatalytic water splitting, the results of long-term chronopotentiometric experiments at 100 and even 500 mA cm^{-2} as well as the post-analysis of the used electrodes and electrolytes should be provided.*

Response: We are grateful to the reviewer for the suggestion. During the revision, we carried out long-term chronopotentiometric tests at high current densities of 100 and 500 mA cm^{-2} . The results have been added to Figure 3f of the revised Manuscript. No significant increase in the overpotential can be observed during the test (duration, 130 h), although the curves are slightly disturbed by the large amount of O_2 bubbles produced by the reaction. Therefore, the high durability of the NiFeB hydroxide catalyst is verified.

We also examined the NiFeB hydroxide nanosheets after the durability test at 500 mA cm^{-2} for 130 h.

Despite the harsh conditions and the long-term application, the sheet-like morphology of the catalyst can still be distinguished by TEM. ICP-MS confirms that the elemental compositions of Ni, Fe, and B in the nanosheets were only slightly changed. All these facts confirm that the catalyst is stable during electrocatalysis. These results have been added to Figures S9, S10, and Table S4 of the revised Supporting Information.

(9) Why when the electrons of Ni are withdrawn by the electron-deficient B proximate to it, the ELF of Ni is enhanced?

Response: We agree with the reviewer that the electron localization function (ELF) is useful for quantifying geographical localization but not the oxidation state caused by B withdrawing electrons from Ni. Therefore, we include the Bader charge analysis that approximates the total electronic charge of an atom by integrating the electron density within Bader volume, which can be used to quantify the cost of removing the charge from an atom. We add the following discussion on page 16 of the revised Manuscript.

“To evaluate the charges quantitatively, we carried out the Bader charge analysis, which shows a maximum of +0.06 in the formal charge of Ni due to the electron withdrawing by B.”

Additionally, we have normalized and replotted ELF for improved clarity. The updated ELFs of the catalysts have been updated in Figure 4b. Related discussions are as follows (page 16):

“Such electronic changes can be indicated by the electron localization functions (ELFs) (Figure 4b). A rainbow color scheme is employed to elucidate the ELFs, in which 0.0 (white) stands for a fully delocalized state and 1.0 (black) stands for a fully localized state. The electron localization of the Ni-bound OH (indicated by the arrows in the image) is strongly enhanced after introducing B, which is indirect evidence of the increasing oxidation state of Ni in the proximity of B.”

(10) It is better to give the loading amounts of Ni, Fe, and B on the electrodes in Table S1.

Response: Thanks for the suggestion. We have updated Table S1 by adding the loading amounts of Ni, Fe, and B on the electrodes.

(11) The very related work on the CoFeB OER catalyst published very recently in Journal of Colloid and Interface Science (DOI:10.1016/j.jcis.2021.07.024) should be cited.

Response: We thank the reviewer for pointing out the reference we overlooked in our previous submission. We have cited this paper in the revised Manuscript.

Reviewer #3: *Bai et al. present a synthesis of single-layer NiFeB hydroxide nanosheets and demonstrate the efficacy of electron-deficient boron in promoting the formation of high-oxidation-state*

Ni for improved OER activity. They reported that a current density of 100 mA cm^{-2} has been achieved in 1.0 M KOH at an overpotential of 252 mV , placing it among the best Ni-based catalysts for the OER.

Response: We thank the reviewer for his/her valuable comments to us. We here respond point by point.

After carefully reading, it looks more like a pretty typical high surface area layered (oxy)hydroxide paper. It's hard to judge how much of the improvement is from fundamental changes in electronic structure vs differences in surface area and morphology. Not that high current isn't important for devices, just more engineering.

Response: The main discovery of this work is that the incorporation of B into a NiFe hydroxide catalyst significantly promotes the oxidation state transition of Ni at a reduced overpotential, leading to a substantial increase in the intrinsic activity of the hydroxide catalyst in the OER. The excellent OER activity is primarily caused by the electronic structure modification of the catalyst, not the high surface area or the specific sheet-like morphology, as commented by the reviewer. We here rationalize our discussion as follows.

The critical role of B in promoting the oxidation state transition of Ni in the hydroxide catalysts has been verified by Raman spectroscopy and X-ray absorption spectroscopy (XAS) (Figure 2). Raman spectroscopy shows that the $\text{Ni}^{2+}(\text{OH})_2 \rightarrow \text{Ni}^{3+\delta}\text{OOH}$ transition occurs at an overpotential of 66 mV in the NiFeB hydroxide catalyst, which is significantly lower than the overpotential required by the B-free NiFe hydroxide catalyst (266 mV). The XAS results also show that the oxidation state transition of Ni occurs at a significantly lower overpotential with the NiFeB hydroxide catalyst, compared with the B-free NiFe hydroxide catalyst, agreeing with the trend observed by the Raman spectroscopy. Both results confirm the critical role of B in promoting the oxidation state transition of Ni in the hydroxide catalysts.

DFT calculations were employed to provide a hint of the electronic interaction mechanism. We carried out the Bader charge analysis, which shows a maximum of $+0.06$ in the formal charge of Ni due to the electron withdrawing by B. Electron localization functions (ELFs) also indicate increasing oxidation state of Ni in proximity of B. These results suggest that the intrinsic electron-withdrawing property of B has contributed to the promoted oxidation state transition of Ni.

The impact of B on the catalytic OER activity of the NiFe hydroxide catalyst was demonstrated experimentally. To disentangle the effect of the B incorporation from that of the surface area on the intrinsic OER activity of the hydroxide catalysts, we calculated the specific activities of the catalysts by normalizing the current densities to the ECSA (Figure 3b). The ECSAs of the hydroxide catalysts were calculated according to the double-layer capacitance of the nanosheets without supporting carbon (Figure S8). Due to the low conductivity of hydroxide catalysts, only a small part of the hydroxide catalysts at their direct contact sites with the electrode was subjected to the analysis. The experimental results show that the intrinsic activity of the NiFeB hydroxide catalyst is 11 times greater than that of the B-free NiFe hydroxide catalyst (Figure 3c), which highlights the critical role of B in achieving

significantly improved OER activity of the NiFeB hydroxide catalyst.

To conclude, all experimental facts confirm the critical role of B in boosting the oxidation state transition of Ni, which is fundamentally important to the electronic engineering of metal hydroxide catalysts toward efficient OER for water splitting. We wish that the value and impact of our findings would be approved by the reviewer.

Moreover, a recent paper (<https://www.nature.com/articles/s41467-021-26307-7>) about NiFe-Borate catalyst just came out. Therefore, the novelty of this study is not high as required by Nature Communications and I don't recommend publishing this paper.

Response: We thank the reviewer for pointing out a recently published paper related to our work. In this paper, the authors report a transformation of NiFe boride into a new NiB₄O₇ phase under the OER potential, which is the active site to promote the *O→*OOH process for improved OER activity. After carefully reading the paper, we conclude that it does not compromise the novelty of our work. The only similarity between our paper and this paper is the elements used for the catalyst fabrication. The two works are different in terms of the material itself and the role of B in the OER catalysis. We here elaborate further on our reasons as follows.

(1) The materials. In our work, we synthesized ultrathin single-layer NiFeB hydroxide nanosheets by *in situ* hydrolysis of NiFeB alloy nanoparticles at ambient conditions. The NiFeB catalyst was composed of a single-layer edge-sharing MO₆ (M = Ni, Fe) octahedra, with BO₃ incorporated into the single layer via B–O–M (M = Ni, Fe) bridging bonds. In the paper raised by the reviewer, the authors synthesized NiFe boride as irregular bulky particles by a chemical reduction method. The NiFe boride transforms into NiFe borate by forming a new phase of NiB₄O₇, which was believed to be the active site for the OER. Therefore, the materials are different in terms of the synthesis method, morphology, structure, and crystal phase.

(2) Role of B in the OER catalysis. In the paper raised by the reviewer, the authors found the new phase of NiB₄O₇ could be the active site for the OER. Although the authors carried out a detailed investigation into the mechanism for the boosted OER by DFT calculations, their focus was on how the new phase affects the elementary steps of the OER. Like other previous works on NiFe boride or borate, the direct role of B in modulating the electronic property of Ni was not clearly revealed. In our work, we discovered the strong effect of B in promoting the oxidation state transition of Ni in the NiFeB hydroxide catalyst by both Raman spectroscopy and X-ray absorption spectroscopy (XAS) techniques. We attributed the effect of B to the unique electron-withdrawing effect of B, as evidenced by DFT calculations. The NiFeB hydroxide nanosheets showed even enhanced OER activity, confirming the efficacy of our catalyst design. Therefore, our study provides the first analysis of the critical role of B in affecting the material itself, i.e., the role in promoting the formation of high-oxidation-state Ni as the active species for the OER. Therefore, our work represents a substantial advance and may guide the design of transition metal oxide (or hydroxide) catalysts for efficient OER toward water splitting.

For the above reasons, we believe our work contains sufficient novelty for its publication in Nature Communications. For better delivering the novelty of this work to readers, we have elaborated on it in the “Discussion” section of the revised Manuscript (page 17).

1. RuO₂ is not a benchmark reference material in 1M KOH.

Response: Thanks for pointing out the inappropriate reference catalyst in our study. Our initial purpose of using RuO₂ as a reference was to make the performance of our catalyst comparable with literature-reported catalysts, because the OER activity of the commercial RuO₂ catalyst has been widely reported in the literature as a reference catalyst even in alkaline media. We agree with the reviewer that RuO₂ is a benchmark OER catalyst in acidic media, not in alkaline media. Transition metal oxides or hydroxides, NiFe layered double hydroxide (NiFe-LDH) for example, show much higher OER activity than the commercial RuO₂. To avoid the confusion caused to the reviewer and other readers, we decide to remove the OER data of RuO₂. Instead, we compare the OER activity of the NiFeB hydroxide nanosheets with the B-free counterpart, which already shows the advantage of B-doping in improving the OER activity of the hydroxide catalyst.

2. In Figure S3, why do Ni 2p and Fe 2p spectra of NiFeB alloy show much noise? While both look much better for NiFeB hydroxide.

Response: Thanks for the comments. The signal-to-noise ratio of the Ni 2p and Fe 2p XPS spectra of the NiFeB alloy nanoparticles depends on the conditions of the XPS analysis, such as the amount of the material subjected to the analysis and the integrity time for the signal acquisition. During the revision, we have collected a new set of spectra with significantly enhanced signal-to-noise ratios. Figure S3 has been updated in the revised Supporting Information.

3. The authors should explain why Ni redox peaks for NiFe hydroxide and NiFeB hydroxide are different. Are they directly related to the Ni valence state change? And how do these Ni redox peaks change when changing the scan rate during OER measurements?

Response: We thank the reviewer for the valuable comments. During the revision, we collected more LSV curves of the NiFeB and NiFe hydroxide catalysts. We found that both the catalysts show similar redox peaks at ~1.45 V. The data have been updated in Figure 3a,c of the revised Manuscript.

According to the reviewer’s suggestion, we have collected the CV curves of the catalysts at different scan rates to examine the characteristics of the peak. The results have been added to Figure S7 of the revised Supporting Information. Generally, the oxidation peak current (I_p) obeys the power-law relation with scan rate (v): $I_p = av^b$. The peak corresponds to a Faradic process if $b = 0.5$ and a charge-discharge process of a double layer capacitor if $b = 1$. We did the fitting and found that b is close to 0.5, suggesting a Faradic process.

According to the Raman and XAS results, the strong, broad peak at 1.39–1.46 V in both the LSV and the CV curves of the NiFeB hydroxide nanosheets may arise from the prominent $\text{Ni}^{2+}(\text{OH})_2 \rightarrow \text{Ni}^{3+\delta}\text{OOH}$ transition in this catalyst. The redox peak of Ni is not obvious for the NiFe hydroxide catalyst, suggesting a low abundance of the active sites in this catalyst. The main redox peak of Ni should appear at a high potential (~ 1.5 V) according to the Raman and XAS results, which may overlap with the OER polarization and become indiscernible. However, we did observe a weak peak at ~ 1.45 V when we zoom in on the LSV curve. This peak can be attributed to sparse $\text{Ni}^{3+\delta}\text{OOH}$ species formed at certain sites of the catalyst prior to the main redox peak of Ni, which, however, should not be the main contributor to the OER activity of the catalyst.

The corresponding discussion has been added to pages 11–12 of the revised Manuscript.

4. Why does introducing B into NiFe only affect the Ni oxidation state? Is there any chance it will also affect the Fe oxidation state?

Response: We thank the reviewer for the comments. The Raman spectroscopy reveals that B promotes the oxidation state transition of Ni in the NiFeB hydroxide nanosheets. It was not clear whether the oxidation state of Fe can be affected by B. To disclose it, we examined the oxidation state of Fe in FeNiB and FeNi hydroxide catalysts by Fe $L_{3,2}$ -edge XAS spectroscopy (Figure S5). For both the catalysts, the spectra show an intense peak ii (Fe $2p \rightarrow 3d e_g$ transition) relative to peak i (Fe $2p \rightarrow 3d t_{2g}$ transition) with an intensity ratio of ~ 2.4 , corresponding to Fe^{3+} in the catalysts. Therefore, we conclude that the incorporation of B does not cause a significant change in the oxidation state of Fe in the hydroxide catalysts.

Further, we examined the oxidation states of Fe in NiFeB and NiFe hydroxide catalysts at different overpotentials. Both XAS spectra display no significant changes with increasing overpotentials, suggesting that the oxidation state of Fe is virtually unchanged during the catalysis.

However, Fe did contribute to the boosted oxidation state transition of Ni in the hydroxide nanosheets. Raman spectroscopy suggests that the overpotential required for the $\text{Ni}^{2+}(\text{OH})_2 \rightarrow \text{Ni}^{3+\delta}\text{OOH}$ transition increased to 166 mV in a Fe-free NiB hydroxide catalyst (Figure S6), which is significantly higher than the overpotential required by a NiFeB hydroxide catalyst observed by the same technique (66 mV, Figure 2a). Therefore, Fe is indispensable in the hydroxide catalysts toward high catalytic efficiencies. We assume that Fe serves as an electron transit for Ni, which boosts the electron transfer at Ni sites but maintains a constant oxidation state by itself.

The results have been added to Figure S5, S6 of the revised Supporting Information and discussed on pages 8–9, 10–11 of the revised Manuscript.

5. In Figure 4b, why does a higher ELF of 1.22 denote a higher Ni oxidation state?

Response: We agree with the reviewer that the electron localization function (ELF) is useful for quantifying geographical localization but not the oxidation state caused by B withdrawing electrons from Ni. Therefore, we include the Bader charge analysis that approximates the total electronic charge of an atom by integrating the electron density within Bader volume, which can be used to quantify the cost of removing the charge from an atom. We add the following discussion on page 16 of the revised Manuscript.

“To evaluate the charges quantitatively, we carried out the Bader charge analysis, which shows a maximum of +0.06 in the formal charge of Ni due to the electron withdrawing by B.”

Additionally, we have normalized and replotted ELF for improved clarity. The updated ELFs of the catalysts have been updated in Figure 4b. Related discussions are as follows (page 16):

“Such electronic changes can be indicated by the electron localization functions (ELFs) (Figure 4b). A rainbow color scheme is employed to elucidate the ELFs, in which 0.0 (white) stands for a fully delocalized state and 1.0 (black) stands for a fully localized state. The electron localization of the Ni-bound OH (indicated by the arrows in the image) is strongly enhanced after introducing B, which is indirect evidence of the increasing oxidation state of Ni in the proximity of B.”

Again, we would like to thank all reviewers for the insightful and constructive comments.

Sincerely,

Chuanbo Gao, PhD
Xi'an Jiaotong University

REVIEWER COMMENTS

Reviewer #1 (Remarks to the Author):

In my opinion the authors have addressed adequately the main comments and criticisms of all reviewers. In particular they provided essential details concerning the choice of the model structure/composition used in the DFT calculations and the related charge analysis that I considered critical in the original submission. The clarifications introduced in the text improved the clarity and readability of the manuscript, which I would consider ready for acceptance.

Reviewer #2 (Remarks to the Author):

In the revised manuscript and the responses, the authors repeatedly emphasize that the high OER activity of NiFeB is attributed to 'the electron-withdrawing role of B in promoting the oxidation state transition of Ni from +2 to +3 and higher'. Such an argument is against the basic chemical concept. The results obtained from Raman and XAS spectra have not been clearly and reasonably explained. To respond my second question, the authors list three examples from the literatures, but these examples do not support their argument. If they carefully read these literatures (especially the paper by Boettcher et al.), they will know that the oxidation potential of Ni²⁺ to Ni³⁺ becomes more difficult (positively shift) with increasing the Fe content in NiFe hydroxides. Furthermore, it is not correct to compare the function of B in NiFeB with that of P in NiCoFeP. As P has two more valence electrons than B, when the P atom is doped in NiCoFe, it should have no apparent electron withdrawing effect.

In addition, the overpotential of an OER catalyst cannot be determined by Raman spectra. The accurate overpotential of an OER catalyst should be determined by LSVs in a proper manner

In Figure 5a, the Raman peaks at ~680 cm⁻¹ is very strong compared with the intensity of peaks from the oxidized Ni species. The authors attributed this peak to the Fe-containing species such as Fe₃O₄, γ-Fe₂O₃, and γ-FeOOH. This causes a doubt that there exist a quite amount of Fe-containing impurities in the as-prepared NiFeB material.

At page 17: "B promotes the oxidation state transition of Ni from the inert Ni²⁺ to the active Ni^{3+δ} at a significantly lower overpotential, which contributes to the boosted OER activity". This conclusive sentence is problematic in chemical concept.

A closely related work on the CoFeB OER catalyst has been published very recently in Journal of Colloid and Interface Science (DOI:10.1016/j.jcis.2021.07.024). Considering the weak innovation of the work and the conceptual problems in the discussion, I cannot give a positive opinion for publication of this work in Nat. Commun.

Reviewer #3 (Remarks to the Author):

The authors have addressed almost all of my concerns in their revised manuscript. The soft XAS results are really helpful and solid to support their argument on the Ni and Fe valence changes in their studied materials. They also highlighted the novelty of this work, which is strongly different from the recently published NC paper. Therefore, now I can recommend accepting it for publication in NC after they correct the following issues.

1. The color of Figure 3f should be revised and keep consistent.
2. To identify the role of Fe in the NiFeB hydroxide, total DOS as well as PDOS should be plotted to study the responding evolution of electronic structure with Fe doped in NiB.

Response to Reviewers

Dear Reviewers,

Thank you very much for offering us the valuable comments and suggestions, which have been very helpful for us to further improve our manuscript. We here respond point by point and submit a revised manuscript with all changes highlighted. We hope you would find that all your concerns have been appropriately addressed.

Response to Reviewer 1

Comment: In my opinion the authors have addressed adequately the main comments and criticisms of all reviewers. In particular they provided essential details concerning the choice of the model structure/composition used in the DFT calculations and the related charge analysis that I considered critical in the original submission. The clarifications introduced in the text improved the clarity and readability of the manuscript, which I would consider ready for acceptance.

Response: We gratefully thank the reviewer for his/her recognition of the value of this work and the valuable comments to us in the previous peer review to help us improve the quality of this work.

Response to Reviewer 2

Comment 1: In the revised manuscript and the responses, the authors repeatedly emphasize that the high OER activity of NiFeB is attributed to “the electron-withdrawing role of B in promoting the oxidation state transition of Ni from +2 to +3 and higher”. Such an argument is against the basic chemical concept. The results obtained from Raman and XAS spectra have not been clearly and reasonably explained. To respond my second question, the authors list three examples from the literatures, but these examples do not support their argument. If they carefully read these literatures (especially the paper by Boettcher et al.), they will know that the oxidation potential of Ni²⁺ to Ni³⁺ becomes more difficult (positively shift) with increasing the Fe content in NiFe hydroxides. Furthermore, it is not correct to compare the function of B in NiFeB with that of P in NiCoFeP. As P has two more valence electrons than B, when the P atom is doped in NiCoFe, it should have no apparent electron withdrawing effect.

Response: We thank the reviewer for the valuable discussion on the oxidation state transition of Ni and the correlated OER activity. We are sorry for the confusion caused to the reviewer by our writing. We here elaborate on it with a major revision to the manuscript. We hope the reviewer would agree with the discussion in the revised form.

We agree with the reviewer that in the conventional NiFe oxyhydroxide catalyst, the oxidation state transition of Ni at a higher overpotential is usually recognized to be favorable for improving the OER. An electron transfer from Ni to Fe makes Ni²⁺ more electron-deficient and thus more difficult to be oxidized, leading to Ni^{3+δ} possessing higher oxidizing power, corresponding to better OER activity.

In this work, we found that the incorporation of B into the NiFe hydroxide catalyst can significantly reduce the potential required for the oxidation state transition of Ni, leading to the formation of active Ni^{3+δ} at a substantially lower potential, which gives rise to excellent OER activity. The same opinion was held by Sargent et al. in *Nat. Chem.* 2018, 10, 149. This opinion is reasonable because OER cannot occur prior to the potential at which the active Ni^{3+δ} species form. Therefore, a significantly reduced potential for the formation of the active Ni^{3+δ} is a prerequisite for achieving extremely high OER activity with a low onset potential. On the other hand, with the facile oxidation state transition of Ni, the active Ni^{3+δ} sites can be formed in a large population. As observed in our experiment (Fig. 3a), the NiFeB hydroxide catalyst shows a significantly larger Ni oxidation peak than the B-free NiFe hydroxide catalyst, contributing to the superior OER activity.

In our previous submission, we attributed the promoted oxidation state transition of Ni to the electron withdrawing effect of B. We agree with the reviewer that this discussion might confuse readers, because it is against the conclusion from the NiFe oxyhydroxide catalytic system. In fact, the seemingly contradictory shift of the redox potential of Ni may have been caused by the different ways of the electron interaction between the Ni²⁺ species and the “electron-withdrawing” component. In the NiFe oxyhydroxide, Fe affects the initial electron density of the Ni²⁺ species, making it more electron-deficient and thus more difficult to be oxidized to Ni^{3+δ}, causing a shift of the redox peak to a higher potential. In our NiFeB hydroxide system, we suppose that the borate species may not significantly change the initial oxidation state of Ni²⁺ but act more as an “electron sink” to facilitate the electron extraction from Ni²⁺ to afford Ni^{3+δ} under an applied potential, thus shifting the redox potential of Ni to a lower overpotential. In fact, a similar phenomenon was observed by Ye et al. (*J. Am. Chem. Soc.* 2016, 138, 9128). In this work, positively-charged plasmonic “holes” in Au nanoparticles promote the oxidation of Ni²⁺ into the active Ni^{3+δ} in Ni(OH)₂, allowing the more efficient water oxidation at lower potentials. From the LSV curve, we can also see a shift of the redox peak of Ni to lower potentials, corresponding to increasing OER activities. This trend agrees well with our NiFeB hydroxide catalyst.

We apologize for supporting the electron withdrawing effect by referring to the work of the NiCoFeP oxyhydroxide (*Nat. Chem.* 2018, 10, 149) in our previous response letter. The authors stated that the incorporation of phosphate promotes the oxidation state transition of Ni but did not specify the electron withdrawing effect. Also, although the paper by Boettcher (*J. Am. Chem. Soc.* 2014, 136,

6744) suggests that the metal oxide/oxyhydroxide→Au electron transfer enhances the OER activity, whether this electron withdrawing effect causes a lower potential for the oxidation state transition of Ni was not discussed in the paper. We have deleted some discussions of the prior works that are inappropriate in the context of the manuscript. We thank the reviewer for correcting our mistake.

To avoid the confusion of the possibly different ways of the electronic interaction in the conventional NiFe oxyhydroxide catalyst and our NiFeB hydroxide catalyst, we have extensively revised the corresponding discussion. We avoid using the term of “electron withdrawing effect” but describe the role of B as an “electron sink” to facilitate the Ni²⁺ oxidation. We hope the reviewer will find that the discussion is more precise in the revised form. Revisions made to the manuscript include:

Page 3: The following discussion was deleted: “In the study of mixed NiFe oxide catalysts, the presence of Fe was found to impose a substantial influence on the Ni⁴⁺/Ni²⁺ ratio^{13,23}. When NiFe hydroxide or NiOOH is formed on an Au substrate as a thin film, its OER activity is significantly improved according to the film thickness, thanks to the partial electron transfer between the catalyst and the Au substrate.¹¹”

Page 4: “We hypothesize that B as an electron-deficient element acts as an electron sink to facilitate the electron extraction from Ni²⁺ to form the desired Ni^{3+δ} at a low overpotential, which is beneficial for the OER²⁹.”

Page 12: “These results suggest that the more facile oxidation state transition of Ni in the NiFeB hydroxide catalyst allows the formation of the active Ni^{3+δ} species not only at a significantly lower overpotential but also in a substantially larger population in the catalyst, which accounts for the superior OER activity.”

Page 17: “The electronic interaction makes it possible for B to impose an electron sink effect on the Ni species to promote its oxidation state transition, leading to a significantly reduced onset potential for the OER and therefore excellent catalytic activity.”

Page 19: “The electron-deficient B may have served as an electron sink for promoting the oxidation of Ni²⁺ into Ni^{3+δ} during the electrocatalytic OER.”

Comment 2: In addition, the overpotential of an OER catalyst cannot be determined by Raman spectra. The accurate overpotential of an OER catalyst should be determined by LSVs in a proper manner.

Response: We agree with the reviewer that the overpotential can be accurately measured by the LSVs. However, the electrochemical measurement cannot provide direct information on the state of the Ni

species, which is an obvious obstacle for the mechanism study. In addition, the redox peak of Ni was not always obvious in the LSV, for example, with the NiFe hydroxide catalyst. Therefore, we examined the evolution of the Ni species at different overpotentials by advanced techniques, including Raman spectroscopy and synchrotron X-ray absorption spectroscopy. Clear information on the oxidation state transition has been revealed by these analyses. The two results have been quite consistent, allowing us to make a decisive conclusion on the critical role of B in promoting the oxidation state transition of Ni and boosting the OER activity.

Following the referee's suggestion, we have removed the Raman-derived specific overpotentials for the oxidation state transition of Ni in the schemes in Figure 2. Instead, we use more qualitative descriptions, i.e., "low overpotential" and "high overpotential". The corresponding quantitative descriptions of the overpotential shift derived from the Raman results were also revised as copied below:

Page 2, Abstract: "Raman spectroscopy and X-ray absorption spectroscopy show that the introduction of B into NiFe hydroxide causes a substantial cathodic shift of the $\text{Ni}^{2+}(\text{OH})_2 \rightarrow \text{Ni}^{3+\delta}\text{OOH}$ transition."

Page 4: "Raman spectroscopy and X-ray absorption spectroscopy show that the transition potential of $\text{Ni}^{2+}(\text{OH})_2$ to $\text{Ni}^{3+\delta}\text{OOH}$ in the NiFeB hydroxide nanosheets is substantially lower than that with the reference B-free NiFe hydroxide nanosheets."

Comment 3: In Figure 5a, the Raman peaks at $\sim 680 \text{ cm}^{-1}$ is very strong compared with the intensity of peaks from the oxidized Ni species. The authors attributed this peak to the Fe-containing species such as Fe_3O_4 , $\gamma\text{-Fe}_2\text{O}_3$, and $\gamma\text{-FeOOH}$. This causes a doubt that there exist a quite amount of Fe-containing impurities in the as-prepared NiFeB material.

Response: We thank the reviewer for the valuable comments. The broad Raman peak at 680 cm^{-1} suggests the presence of Fe–O, not Fe-containing impurities, in the NiFeB hydroxide catalyst. Such a Raman peak can also be observed in Fe oxides or oxyhydroxides, such as Fe_3O_4 , $\gamma\text{-Fe}_2\text{O}_3$, and $\gamma\text{-FeOOH}$ which contain the same structural unit. We are sorry for the misleading statement in our previous submission, which has been revised as copied below to make it precise.

Page 8: "The broad Raman peak at 680 cm^{-1} can be attributed to the Fe–O in the NiFeB hydroxide nanosheets, as this peak can also be observed in Fe_3O_4 , $\gamma\text{-Fe}_2\text{O}_3$, and $\gamma\text{-FeOOH}$ that contain the same structural unit."

Comment 4: At page 17: “B promotes the oxidation state transition of Ni from the inert Ni²⁺ to the active Ni^{3+δ} at a significantly lower overpotential, which contributes to the boosted OER activity” . This conclusive sentence is problematic in chemical concept.

Response: We thank the reviewer for the valuable comment. Following the reviewer’s suggestion, we have reworded this discussion as follows, according to our response to Comment 1:

Page 18-19: “The incorporation of B in the NiFeB hydroxide reduces the potential required for the oxidation of Ni²⁺ into the active Ni^{3+δ} species, which contributes to the decreased onset potential and high activity of the OER.”

Comment 5: A closely related work on the CoFeB OER catalyst has been published very recently in Journal of Colloid and Interface Science (DOI: 10.1016/j.jcis.2021.07.024). Considering the weak innovation of the work and the conceptual problems in the discussion, I cannot give a positive opinion for publication of this work in *Nat. Commun.*

Response: Thanks for the discussion on the novelty of this work. In the paper raised by the reviewer, a Co-Fe-B ternary catalyst with improved crystallinity was synthesized by chemical reduction and subsequent calcination. XAS and XPS results show that the redistribution of electronic structures is beneficial for improving the OER performance. After carefully reading this paper, we conclude that this work is different and will not compromise the novelty of our work.

(1) The materials are different. In our work, we synthesized ultrathin single-layer (~0.5 nm) NiFeB hydroxide nanosheets by in situ hydrolysis of NiFeB alloy nanoparticles at ambient conditions. The NiFeB catalyst was composed of a single-layer edge-sharing MO₆ (M = Ni, Fe) octahedra, with BO₃ incorporated into the single layer via B–O–M (M = Ni, Fe) bridging bonds. In the paper raised by the reviewer, the authors synthesized Co-Fe-B ternary catalyst as irregular bulky particles by chemical reduction followed by calcination. The materials are different in terms of the synthesis method, chemical composition, morphology, and crystal structure.

(2) Role of B in the OER catalysis. In the paper raised by the reviewer, the authors found an electron transfer from Fe to B in the Co-Fe-B catalyst. However, how this electron transfer affects the OER activity and whether this electron transfer imposes any influence on the active Co centers remain unexplored. Our work dedicates to revealing these two critical questions. We discovered for the first time the strong effect of B in promoting the oxidation state transition of Ni in the NiFeB hydroxide catalyst by both Raman spectroscopy and XAS. We attribute the effect of B to the unique electron deficiency of B according to DFT calculations. Guided by these understandings, we can achieve much more superior OER activity. Therefore, our work represents a substantial advance in understanding and designing transition metal-based catalysts for efficient OER.

Therefore, our work is novel and different from the reference paper. We believe our work represents a significant advance in understanding and designing transition metal-based catalysts for the OER toward efficient water splitting.

Response to Reviewer 3

Comment 1: The authors have addressed almost all of my concerns in their revised manuscript. The soft XAS results are really helpful and solid to support their argument on the Ni and Fe valence changes in their studied materials. They also highlighted the novelty of this work, which is strongly different from the recently published NC paper. Therefore, now I can recommend accepting it for publication in NC after they correct the following issues.

Response: We gratefully thank the reviewer for his/her recognition of the value of this work and the valuable comments to us. We here respond point by point.

Comment 2: The color of Figure 3f should be revised and keep consistent.

Response: We are sorry for the inconsistent color legend in Fig. 3f, which has been corrected in the revised manuscript. We thank the reviewer for the kind advice.

Comment 3: To identify the role of Fe in the NiFeB hydroxide, total DOS as well as PDOS should be plotted to study the responding evolution of electronic structure with Fe doped in NiB.

Response: We agree with the reviewer that DOS can provide electronic structure information, which is helpful to identify the role of Fe in the electrochemical reactions. Therefore, we have added the DOS and pDOS to Figure S12 of the revised Supporting Information. The following discussion is also added to the revised manuscript:

Page 18: “The roles of Ni and Fe in the NiFeB hydroxide were further investigated by the density of state (DOS) analysis and a projected DOS (*p*DOS) analysis (Figure S12). It appears that the electronic states near the Fermi level are predominantly from Ni and Fe 3*d* orbitals. This analysis indicates that both Ni and Fe can act as catalytic active sites by contributing the states near the valance band-edge. Indeed, such a conclusion is consistent with our proposed mechanism, in which Fe provides the adsorption site for *O and Ni is the adsorption site for other OER intermediates (Figure 4a, inset).”

Again, we would like to thank all the reviewers for their insightful comments, which have been very helpful for us to further improve the quality of this work. We thank all the reviewers for their valuable time in reviewing our manuscript.

Sincerely,

Chuanbo Gao
Xi'an Jiaotong University

REVIEWER COMMENTS

Reviewer #2 (Remarks to the Author):

The authors have adequately addressed most of my comments, and the revised manuscript would be suitable for publication in Nat. Commun. after a further minor revision for the following problems.

(1) The most important opinion in this manuscript is that "B as an electron-deficient element acts as an electron sink to facilitate the electron extraction from Ni²⁺ to form the desired Ni^{3+δ} at a low overpotential, which is beneficial for the OER." Please give a more clear explanation for this argument to avoid the misunderstanding of the audience.

(2) In the revised manuscript, the authors often use "significantly lower" and "substantially lower" to describe the changes of the oxidation potentials and OER onset potentials, such as "at a significantly lower overpotential", "a significantly reduced onset potential", "is significantly higher than", "a substantial cathodic shift", "substantially lower than", "decreases significantly to", "leading to a significantly reduced onset potential". Such descriptions are too vague. To be objective and accurate, it is better to show the changes of the potentials and overpotentials by using experimental data from electrochemical tests.

(3) I insist that "the overpotential of an OER catalyst cannot be determined by Raman spectra". If the redox peak of Ni of the samples cannot be clearly detected by LSVs, I suggest that the authors could measure the DPVs of the samples to get the wanted redox data and discuss the changes of the potentials and overpotentials by using these accurate data.

Response to Reviewers

Dear Reviewers,

Thank you very much for offering us the valuable comments and suggestions, which have been very helpful for us to improve our manuscript. We here respond point by point and submit a revised manuscript with all changes highlighted. We hope you would find that all your concerns have been appropriately addressed.

Response to Reviewer 2

The authors have adequately addressed most of my comments, and the revised manuscript would be suitable for publication in Nat. Commun. after a further minor revision for the following problems.

Response: We gratefully thank the reviewer for his/her approval of our previous revision and the valuable comments to us.

Comment 1: The most important opinion in this manuscript is that “B as an electron-deficient element acts as an electron sink to facilitate the electron extraction from Ni^{2+} to form the desired $\text{Ni}^{3+\delta}$ at a low overpotential, which is beneficial for the OER.” Please give a more clear explanation for this argument to avoid the misunderstanding of the audience.

Response: We thank the reviewer for the valuable suggestion. We have made more explanations on the electron sink effect of B in our revised manuscript (copied below). We hope that our revised discussion becomes clearer to readers.

Page 4: “Our perspective to address this issue is to introduce the electron-deficient boron (B) into the conventional NiFe hydroxide catalysts for the OER. Usually, directly extracting electrons from Ni^{2+} to form the desired $\text{Ni}^{3+\delta}$ is a difficult process and can be achieved at a high potential. We expect that the B at the proximity of Ni may participate in the electron extraction process by serving as a transit site for the electron flow. Owing to the intrinsic electron deficiency, the transit B site may act as an “electron sink” to facilitate the electron flow from the Ni^{2+} site, thus allowing for the formation of the active $\text{Ni}^{3+\delta}$ at a reduced potential, which is beneficial for the OER²⁹.”

Comment 2: In the revised manuscript, the authors often use “significantly lower” and “substantially lower” to describe the changes of the oxidation potentials and OER onset potentials, such as “at a significantly lower overpotential”, “a significantly reduced onset potential”, “is significantly higher than”, “a substantial cathodic shift”, “substantially lower than”, “decreases significantly to”, “leading

to a significantly reduced onset potential”. Such descriptions are too vague. To be objective and accurate, it is better to show the changes of the potentials and overpotentials by using experimental data from electrochemical tests.

Response: We thank the reviewer for the valuable comments, which we fully agree with the reviewer. We have removed all these subjective discussions from the revised manuscript. Following the referee’s suggestion, we used experimental data from electrochemical tests to describe the changes of the potentials for the oxidation state transition of Ni. We include the following statement in the Discussion section:

Page 18: “More specifically, DPV analysis show that the incorporation of B into the NiFe hydroxide catalyst has caused a 60-mV cathodic shift of the potential required to initiate the oxidation state transition of Ni.”

Comment 3: I insist that “the overpotential of an OER catalyst cannot be determined by Raman spectra”. If the redox peak of Ni of the samples cannot be clearly detected by LSVs, I suggest that the authors could measure the DPVs of the samples to get the wanted redox data and discuss the changes of the potentials and overpotentials by using these accurate data.

Response: We thank the reviewer for the valuable suggestion. Following his/her suggestion, we have measured the differential pulse voltammetry (DPV) of our samples. The DPV results show that the oxidation state transition of Ni in the NiFeB hydroxide catalyst appear at a lower potential, compared with the B-free NiFe hydroxide catalyst, which is generally consistent with the Raman and XAS observations. The DPVs of the NiFeB and NiFe hydroxide nanosheets have been added to **Figure 3d** of the revised manuscript. More discussion has been made as copied below:

Page 12: “The $\text{Ni}^{2+}(\text{OH})_2 \rightarrow \text{Ni}^{3+\delta}\text{OOH}$ transition was further investigated electrochemically by investigating the Ni oxidation peak by differential pulse voltammetry (DPV) (Figure 3d). The onset potentials of the Ni oxidation peaks were 1.33 and 1.39 V vs. RHE with the NiFeB and NiFe hydroxide nanosheets, respectively. Therefore, the incorporation of B into the NiFe hydroxide catalyst has caused a 60-mV cathodic shift of the potential required to initiate the oxidation state transition of Ni. The peak position also underwent a cathodic shift by 20 mV with B. This result again highlights the efficacy of B in promoting the formation of high-oxidation-state Ni species at low overpotentials, consistent with the Raman and XAS observations.”

In addition, we made minor revisions to the Abstract and some other parts of the manuscript to make our discussion more accurate. In order to examine the onset potential of the OER more precisely, we

also performed cathodic LSV. The results have been added to Figure S7 and discussed in the revised manuscript on page 12.

Again, we would like to thank the reviewer for his/her insightful comments, which have been very helpful for us to improve the quality of this work. We thank the reviewer for his/her valuable time in reviewing our manuscript.

Sincerely,

Chuanbo Gao
Xi'an Jiaotong University

REVIEWERS' COMMENTS

Reviewer #2 (Remarks to the Author):

All my concerns have been well addressed after the third round of revision. In my opinion, the manuscript could be published in Nat. Commun. as it stands.

Response to Referees

REVIEWERS' COMMENTS

Reviewer #2 (Remarks to the Author):

All my concerns have been well addressed after the third round of revision. In my opinion, the manuscript could be published in Nat. Commun. as it stands.

Response: We appreciate the referee's agreement on our previous revision.

Again, we would like to thank all the referees for their valuable comments on our manuscript, which have been greatly helpful to improve the quality of this work.